# A Closer Look at In-context Learning of LLMs in Simple Classification Tasks

## Abstract

*In-context learning* (ICL), as an emergent behavior of large language models (LLMs), has exhibited impressive capability in solving previously unseen tasks through observing several given in-context examples without further training. However, recent works find that LLMs irregularly obtain unexpected fragmented decision boundaries in simple machine learning classification tasks (e.g., binary linear classification). Although some efforts have been made in this problem, the phenomenon remains under-explored. Thus, in this paper, we first explore the in-context learning capability of LLMs with both implicit and explicit reasoning paradigms. Our observations on the behaviors of LLMs indicate that LLMs consistently fail to achieve smooth decision boundaries in all cases and implicit reasoning is able to achieve better decision boundaries than explicit reasoning. Moreover, LLMs tend to address classification tasks in the way of machine learning algorithms. With these basic observations, we propose to dive into the behaviors of LLMs for a deeper understanding of their in-context learning capability on discriminative tasks. To this end, we conduct a series of analyses on LLMs to explore how LLMs perform discriminative tasks. We explore the behaviors of LLMs in performing classification by prompting LLMs with specified machine learning algorithms and in high-dimensional classification tasks. Then, we propose a method to determine whether LLMs implicitly leverage machine learning algorithms when addressing classification tasks. Moreover, we also rethink the decision boundaries of LLMs from the perspective of data distributions. Overall, our analyses provide important observations and insights into the behaviors of LLMs in the discriminative tasks.

## 1 Introduction

Large language models (LLMs) (e.g., Llama (Touvron et al., 2023), Qwen (Yang et al., 2024a)), which are equipped with billions of parameters and pre-trained on huge amounts of corpora, have exhibited impressive capability in solving various kinds of tasks, ranging from reasoning commonsense to solving arithmetic problems (Lewkowycz et al., 2022; Wei et al., 2022; Kojima et al., 2022; Suzgun et al., 2022). The impressive achievements, to some extent, should be attributed to a significant ability, which is known as in-context learning (a.k.a. ICL) (Brown, 2020), derived from those large-scale transformer-based language models. Specifically, in the in-context learning problem, with elaborately designed prompts, LLMs can be adapted to previously unseen tasks conditioned on several given in-context examples and instructions (Wei et al., 2022) without having to be further explicitly trained.

Although in-context learning plays an important role in many tasks, the underlying mechanism of this learning paradigm remains unclear. Thus, an essential research topic in in-context learning is determining the underlying mechanism of in-context learning so that a comprehensive understanding of such a powerful learning paradigm can be obtained. So far, many works have been done from both theoretical and empirical perspectives (Von Oswald et al., 2023; Dai et al., 2023; Shi et al., 2023; Wei et al., 2023; Webson and Pavlick, 2021; Chen et al., 2024; Reid et al., 2024; Agarwal et al., 2024; Bertsch et al., 2024; Garg et al., 2022; Nguyen and Grover, 2022). In particular, Von Oswald et al. (2023) and Dai et al. (2023) find that the underlying mechanism of the in-context learning resembles gradient descent, which implies that transformers can emulate the optimization processes. Inspired by this finding, Zhao et al. (2024) recently proposes to qualitatively analyze the in-context learning paradigm through the decision boundaries generated by LLMs (e.g., Llama and ChatGPT) on both linear and non-linear machine learning classification tasks (e.g., the simple binary classification task).

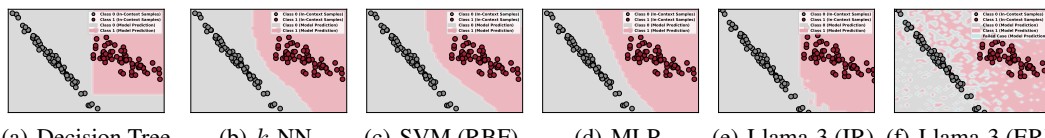

(a) Decision Tree    (b) $k$-NN    (c) SVM (RBF)    (d) MLP    (e) Llama-3 (IR)    (f) Llama-3 (ER)

Figure 1: **A review of decision boundaries of both conventional machine learning algorithms and LLMs (take Llama-3-8B-Instruct as an example) on the binary classification task with implicit and explicit reasoning.** Figs. (a)-(d) show the decision boundaries of conventional machine learning algorithms, including Decision Tree, $k$-NN, SVM, and MLP. Fig. (e) shows the decision boundary derived from Llama-3-8B via implicit reasoning. Fig. (f) shows the decision boundary derived from Llama-3-8B via explicit reasoning. According to the visualization results, it is easy to observe that the decision boundaries of conventional machine learning methods are smooth and clear, while the decision boundaries of Llama-3-8B (both implicit and explicit reasoning) are fragmented.

Although great success has been achieved in the complex problems (Achiam et al., 2023; Lampinen et al., 2022; Suzgun et al., 2022; Yang et al., 2024b; Wei et al., 2022), Zhao et al. (2024) finds that LLMs perform poorly on simple binary classification tasks. Specifically, compared to conventional machine learning algorithms (e.g., SVM and MLP), which can generate smooth decision boundaries, LLMs merely obtain fragmented decision boundaries on the linearly separable data (see Fig. 1). Such a phenomenon indicates that LLMs are not competent enough to infer the labels of query data by merely observing the in-context data. Although a series of analyses, ranging from the hyperparameter settings (e.g., the numbers of model parameters and in-context data) to prior knowledge (e.g., the order of in-context data and the format of labels), have been done by Zhao et al. (2024), such a phenomenon remains under-explored. Thus, in this work, we propose to dive into the behavior of LLMs in classification tasks to achieve a deeper understanding of their in-context learning capability.

In this paper, we first evaluate the performance of LLMs on the linear classification task with both implicit and explicit reasoning paradigms (see decision boundaries of Llama-3-8B in Figs. 1(e)-1(f)). The main difference between implicit and explicit reasoning is whether the answer is inferred from explicit logical analyses. Specifically, implicit reasoning requires LLMs to output the answer directly, while explicit reasoning requires LLMs to perform the inference with detailed analyses (e.g., CoT (Wei et al., 2022)). Through examining the output of explicit reasoning, we observe a phenomenon that LLMs consistently infer the labels through analyzing the pattern of the data. Moreover, some LLMs, such as Llama-3-8B, may try to leverage existing machine learning algorithms, such as $k$-NN, and other statistical methods (e.g., the mean and standard deviation). These phenomena indicate that LLMs tend to infer the label of query data classification tasks in a similar way to machine learning.

With these basic observations, we propose to dive into the behaviors of LLMs for a deeper understanding of their in-context capability in discriminative tasks. To this end, we first propose a new metric, *smooth score*, to quantitatively measure the fragmentedness of decision boundaries. Then, we explore the behaviors of LLMs in performing classification tasks with prompts where learning strategies are specified. Meanwhile, for explicit reasoning, we also explore the effect of robust reasoning (e.g., SC-CoT (Wang et al., 2023)), since a single reasoning step may include incorrect calculations and executions of learning strategies. Moreover, the introduction of the smooth score further extends the study scope of decision boundaries to high dimensions. Thus, we also determine the behavior of LLMs in high-dimensional cases (e.g., 3D classification tasks). Among all cases, we find that implicit reasoning without specified learning strategies consistently achieves smoother decision boundaries.

The phenomenon that implicit reasoning obtains better decision boundaries than explicit reasoning implies that LLMs intrinsically possess the basic capability in solving classification tasks and extra instructions may damage the capability. However, due to the black-box nature of the implicit reasoning, such a capability remains unclear. As a solution, we propose to measure the similarities of distributions between the predictions of LLMs and a potential strategy. The insight here is that the distribution of the predictions of LLMs should be similar to that of the predictions derived from the strategy if a specific strategy is (implicitly) adopted by LLMs. In this work, we mainly consider the machine learning methods (conducted by scikit-learn (Pedregosa et al., 2011)). Meanwhile, in the previous work, the query data are derived from the plane, where the in-context data lie. Actually, from the perspective of data distribution, the query data generated in the typical way contain both in-distribution and out-of-distribution data. This further motivates us to rethink the decision boundaries of LLMs from the perspective of the distribution discrepancies between in-context and query data.

Overall, our contribution in this paper can be primarily summarized as the following several aspects:

- In Section 3, we review the decision boundaries of LLMs on the linear classification task with both implicit and explicit reasoning. We find that the decision boundaries of both cases are fragmented, while the decision boundaries of implicit reasoning are better (Section 3.1). Moreover, we furt find that LLMs tend to perform the classification task in the way of machine learning (Section 3.2).
- In Section 4, we first propose a new metric, *smooth score*, to quantitatively describe the fragmentedness of decision boundaries (Section 4.1). Then, we explore the behaviors of LLMs in performing classification with prompts in which learning strategies are specified and the behaviors of LLMs equipped with robust reasoning paradigms (Section 4.2 & 4.3). Moreover, we also further extend our study of decision boundaries to high dimensions with the proposed smooth score (Section **??**).
- In Section 5, we rethink decision boundaries of LLMs from the perspective of data distributions. We first propose a new method to determine the relation between the behaviors of LLMs and the predictions of conventional machine learning algorithms when performing classification tasks by measuring the difference between the distributions of the decision boundaries of LLMs and conventional ML algorithms (Section 5.1). Then, we explore whether the decision boundaries of LLMs are affected by the distribution discrepancies between in-context and query data (Section 5.2).

## 2 PROBLEM FORMULATION

**ICL Formulation.** Consider a pretrained large language model parameterized with $\theta^*$, which is pretrained on a large amount of corpus, and a set of labeled data $\mathcal{D}_{\text{IC}} = \{(\boldsymbol{x}_i^{\text{IC}}, y_i^{\text{IC}})\}_{i=1}^{|\mathcal{D}_{\text{IC}}|}$, $y_i^{\text{IC}} \in \{0, 1, ..., N_{\text{cls}} - 1\}$, where $\boldsymbol{x}_i^{\text{IC}} \in \mathbb{R}^d$ and $y_i^{\text{IC}}$ respectively denote the $i$-th $d$-dimension data point and its corresponding label, and $N_{\text{cls}}$ denotes the number of classes in $\mathcal{D}_{\text{IC}}$. Then, given a previously unseen query data point $\boldsymbol{x}^{\text{query}} \in \mathbb{R}^d$, the label of the query data point can be predicted via conditioning on the data in $\mathcal{D}_{\text{IC}}$. To be specific, the inference of the label can be formulated as:

$$P(\hat{y}^{\text{query}}|\boldsymbol{x}^{\text{query}}, (\boldsymbol{x}_1^{\text{IC}}, y_1^{\text{IC}}), ..., (\boldsymbol{x}_{|\mathcal{D}_{\text{IC}}|}^{\text{IC}}, y_{|\mathcal{D}_{\text{IC}}|}^{\text{IC}}), \theta^*). \tag{1}$$

Problem (1) is also known as *in-context learning*. In in-context learning, LLMs are allowed to infer the labels of query data based on several labeled data samples that partially reflect the characteristics of the tasks. Here, $\mathcal{D}_{\text{IC}}$ is known as the in-context data set in the setting of the in-context learning.

**Task Formulation.** In this paper, we mainly probe the in-context learning capability of LLMs through simple classification tasks, such as the linear classification task, which are widely adopted in conventional machine learning works. The generation of tasks and the hyperparameter settings are consistent with Zhao et al. (2024). More details regarding task settings are available in Appendix B.

Consider a set of in-context samples $\mathcal{D}_{\text{IC}} = \{(\boldsymbol{x}_i^{\text{IC}}, y_i^{\text{IC}})\}_{i=1}^{|\mathcal{D}_{\text{IC}}|}$, which is composed of data from $N_{\text{cls}}$ classes. We assume that each data pair $(\boldsymbol{x}^{\text{IC}}, y^{\text{IC}}) \in \mathcal{D}_{\text{IC}}$ are uniformly sampled from a distribution $p_{\text{data}}$. Then, we can respectively obtain the minimum and maximum values $\boldsymbol{x}_{\min}^k \in \mathbb{R}$, $\boldsymbol{x}_{\max}^k \in \mathbb{R}$ along *each dimension* $k \in \{1, 2, ..., d\}$ of the data. Next, we uniformly divide each dimension into $N_{\text{g}}$ coordinates. Specifically, the $j$-th coordinate of dimension $i$ can be expressed as $c_j^i = \boldsymbol{x}_{\min}^i + \frac{j}{N_{\text{g}}}(\boldsymbol{x}_{\max}^i - \boldsymbol{x}_{\min}^i)$. In such a case, a set of $N_{\text{g}}^d$ points can be obtained by combining these coordinates from different dimensions. For example, when $N_{\text{g}}$ and $d$ are respectively set to 50 and 2, we can obtain 2500 two-dimensional query data, which are uniformly distributed in a plane.

In this paper, the aforementioned $N_{\text{g}}^d$ synthetic data are treated as the query data of each classification task, which is denoted by $\mathcal{D}^{\text{query}} = \{(\boldsymbol{x}_i^{\text{query}}, y_i^{\text{query}})\}_{i=1}^{N_{\text{g}}^d}$. In this paper, the LLMs are expected to infer the labels of these query data based on the observation of the in-context data $\mathcal{D}_{\text{IC}}$ via Eq. (1).

## 3 REVISIT DECISION BOUNDARIES OF IN-CONTEXT LEARNING IN LLMS

In this section, we propose to revisit the decision boundaries of in-context learning in large language models. The original work of exploring the decision boundaries of in-context learning in LLMs is conducted by Zhao et al. (2024). Compared to conventional reasoning tasks (e.g., math reasoning tasks), conventional machine learning classification tasks are more challenging to LLMs due to the implicit task pattern. In the original work, LLMs are required to *directly* perform inference to predict the labels of query data that are uniformly distributed in a plane. Such an inference paradigm is also known as *implicit reasoning* (IR) (Deng et al., 2024). However, LLMs are also famous for their

Table 1: Examples of explicit reasoning responses of LLMs (Llama-3-8B and Mistral-8B).

| Model | Explicit Reasoning Responses |
|---|---|
| Llama-3-8B | To predict the label for the new input pair, I will analyze the provided data and identify the patterns and relationships between the input features and the corresponding labels. First, I will calculate the mean and standard deviation of the input features to get an idea of the scale and distribution of the data. Mean of input features: -0.011. Standard deviation of input features: 1.4354. Next, I will calculate the correlation coefficient between the input features and the labels to identify the strength and direction of the relationship. Correlation coefficient: 0.73 The correlation coefficient indicates a moderate positive correlation between the input features and the labels. This suggests that the input features are related to the labels, but there may be some noise or outliers in the data. Now, I will use the **k-nearest neighbors (KNN)** algorithm to predict the label for the new input pair. I will use the Euclidean distance metric to calculate the distance between the new input and the existing data points. K = 5 (number of nearest neighbors). Distance calculation: \| -3.1663732465591417 - 2.1063395598450043 \| = 5.272712806404145. The 5 nearest neighbors to the new input are: 1. Input: -3.5978442841503377 3.970694307972864, Label: 0 2. Input: -3.028945019911411 3.0487289716533956, Label: 0 3. Input: -2.8462232000301673 3.1313644971154407, Label: 0 4. Input: -2.974615433260139 3.2671021888452083, Label: 0 5. Input: -3.3208954438134493 3.746249863497515, Label: 0 The majority of the nearest neighbors have a label of 0, so I will predict the label for the input as 0. Confidence: 0.8 |
| Mistral-8B | ### Reasoning Process: 1. Observation of Patterns: From the provided data, we can observe that pairs with a negative first number and a positive second number are labeled as 0. This is a consistent pattern in the dataset. 2. Application to New Input: The new input pair (-3.5978442841503377, -0.8373784635147727) fits this pattern, as the first number is negative and the second number is also negative. 3. Confidence: Given the consistent pattern observed, I am confident that the label for this input pair is 0. |

impressive capability in reasoning answers in a step-by-step way (i.e., CoT). To make it clear, the latter reasoning paradigm is named as *explicit reasoning* (ER) in this paper. In this section, to obtain a comprehensive understanding of the decision boundaries of in-context learning in LLMs, we propose to conduct in-context learning via both implicit and explicit reasoning on simple classification tasks.

## 3.1 OBSERVATIONS ON DECISION BOUNDARIES OF LLMs

To obtain an understanding of the behaviors of LLMs in ICL, we propose to perform the inference on the linear classification task with both implicit and explicit reasoning. Compared to the non-linear tasks, the linear classification task is easier since the data clusters in the task are linearly separable. The results of the decision boundaries of both conventional machine learning algorithms and LLMs are visualized in Fig. 1. For simplicity of presentation, we take Llama-3-8B as an example of LLMs.

***Observation 3.1.*** **The decision boundaries obtained from LLMs are fragmented.** Figs. 1(a) - 1(d) visualize the decision boundaries derived from conventional machine learning algorithms (Decision Tree, $k$-NN, SVM, and MLP) while Figs. 1(e) - 1(f) visualize the decision boundaries derived from Llama-3-8B. We can observe that the decision boundaries of conventional machine learning methods are smooth and clear, while the decision boundary of Llama-3-8B is fragmented. Meanwhile, according to Zhao et al. (2024), the fragmented boundary cannot be improved by simply modifying hyperparameter settings (e.g., the model size) and prior knowledge (e.g., the label orders/names).

***Observation 3.2.*** **The decision boundary derived from implicit reasoning is better than that from explicit reasoning.** As aforementioned, the main difference between implicit and explicit reasoning is whether the inference is performed in a step-by-step thinking way. Intuitively, due to the impressive capability of explicit reasoning in typical mathematical tasks, the boundary derived from explicit reasoning is expected to be better than that from implicit reasoning. However, according to Fig. 1(f), the decision boundary with explicit reasoning is more fragmented than that with implicit reasoning.

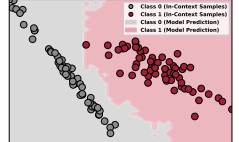 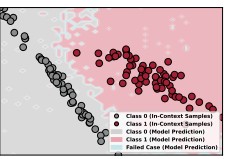

(a) DeepSeek-V3 (IR)  (b) DeepSeek-V3 (ER)

Figure 2: Decision boundaries on DeepSeek-V3 with both implicit and explicit reasoning.

To demonstrate the generality of the two observations obtained above, we also perform both implicit and explicit reasoning on DeepSeek-V3 (Liu et al., 2024) with the same settings. According to Fig. 2, we can observe that the decision boundaries derived from the much larger LLM remain fragmented. Meanwhile, the decision boundary derived from implicit reasoning is consistently much better than that derived from explicit reasoning. In addition, we also evaluate Mistral-8B (Jiang et al., 2023) and Qwen-2.5-7B (Yang et al., 2024a) on the linear classification task. The results are visualized in Fig. 10 (cf. Appendix C), which demonstrates that the two observations are consistent in these LLMs.

Table 2: Smooth scores of decision boundaries of both conventional ML methods (scikit-learn) and implicit reasoning of LLMs prompted with specified machine learning methods (Specified IR).

| Case | Standard | Any ML | Decision Tree | $k$-NN | MLP | SVM |
|---|---|---|---|---|---|---|
| Baseline (sklearn) | - | - | 0.9868 | 0.9854 | 0.9854 | 0.9830 |
| Llama-3-8B (Specified IR) | 0.9772 | 0.9764 | 0.9768 | 0.9758 | 0.9758 | 0.9750 |

## 3.2 OBSERVATIONS ON EXPLICIT REASONING RESPONSES OF LLMS

As a follow-up step to explore the behaviors of LLMs in classification tasks with in-context learning, we propose to probe the reasoning process of LLMs. However, due to the black-box nature of LLMs, it is intractable to directly probe the implicit reasoning process. Thus, in this part, we mainly focus on explicit reasoning. Two response examples of Llama-3-8B and Mistral-8B are presented in Table 1.

***Observation 3.3.* LLMs tend to address the classification task in the way of machine learning algorithms.** According to the responses shown in the table, both LLMs tend to first analyze the patterns of in-context data. Then, the predictions are performed based on the patterns (see the response of Mistral-8B) or by leveraging the machine learning algorithms (see the response of Llama-3-8B).

## 4 EXPLORATION ON IMPLICIT AND EXPLICIT REASONING OF LLMS

In the previous section, we initially probe the behaviors of LLMs in addressing ML classification tasks with in-context learning. In this section, as a further step, we propose to explore the characteristics of implicit and explicit reasoning paradigms of LLMs in performing the linear classification task. Complete discussions including more LLMs and other classification tasks are available in Appendix D.

## 4.1 SMOOTH SCORE: MEASURE THE FRAGMENTEDNESS OF DECISION BOUNDARIES

As far as we know, existing work (Zhao et al., 2024) only qualitatively describes the fragmented decision boundaries, and there isn't any quantitative metric for such a phenomenon. This constrains the exploration of decision boundaries of LLMs in higher dimensions since visualizing the decision boundaries in high dimensions (e.g., 3-dimensional case) is intractable. To solve this problem, we propose a metric, *smooth score*, to quantitatively measure how fragmented decision boundaries are.

**Definition 4.1** (Smooth Score). *Given a set of query data $\mathcal{D}^{\text{query}} = \{(\boldsymbol{x}_i, \hat{y}_i)\}_{i=1}^{N_{\text{query}}}$, where $N_{\text{query}}$ denotes the number of query data, $\boldsymbol{x}_i \in \mathbb{R}^d$ denotes the $i$-th query sample, and $\hat{y}_i \in \{0, 1, ..., N_{\text{cls}}-1\}$ denotes the corresponding prediction of its label. For an arbitrary query data pair $(\boldsymbol{x}, \hat{y}) \in \mathcal{D}^{\text{query}}$, there exist $K$ nearest neighbors $\mathcal{C}^{\text{KNN}} = \{(\boldsymbol{x}_j', \hat{y}_j')\}_{j=1}^K$. The smooth score $\mathcal{S}_{\mathcal{D}^{\text{query}}}$ is calculated as:*

$$\mathcal{S}_{\mathcal{D}^{\text{query}}} = \frac{1}{N_{\text{query}} \cdot K} \sum_{(\boldsymbol{x}, \hat{y}) \in \mathcal{D}^{\text{query}}} \sum_{(\boldsymbol{x}', \hat{y}') \in \mathcal{C}^{\text{KNN}}} \mathbb{1}(\hat{y}' = \hat{y}). \tag{2}$$

The smooth score measures the smoothness of decision boundaries by determining the number of data points that have different prediction results in a small neighborhood around a given sample. Intuitively, in a case where a smooth decision boundary exists, each query sample (except for the query samples at the decision boundary) shares the same prediction as other samples in the vicinity.

In the context of this paper, two main properties of our proposed smooth score can be summarized: (1) the smooth score is loosely lower and upper bounded by 0 and 1: $\mathcal{S} \in (0, 1)$; (2) the smooth score increases with the improvement of the smoothness of the decision boundaries. In addition, more analytical results regarding the robustness of our proposed smooth score are reported in Appendix E.

## 4.2 DOES IMPLICIT REASONING BENEFIT FROM SPECIFIED ML STRATEGIES?

In the previous section, we observe that LLMs with implicit reasoning achieve better decision boundaries in the classification tasks (Figs. 1(e) & 10(a)), and LLMs tend to handle classification tasks via data pattern analyses or machine learning methods when explicit reasoning is performed. The two observations motivate us to explore whether LLMs benefit from specifying machine learning methods in prompts. Thus, in this section, we propose to have LLMs perform implicit reasoning on the linear classification task with prompts, where learning strategies are specified. The specified learning strategies include any machine learning methods, Decision Tree, $k$-NN, MLP, and SVM.

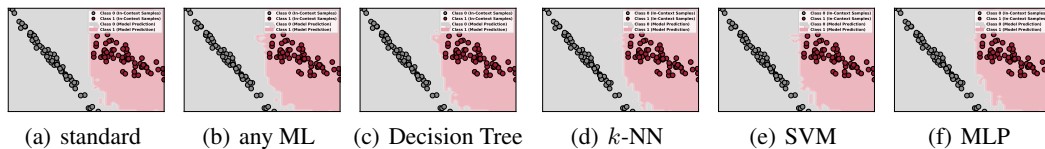

(a) standard  (b) any ML  (c) Decision Tree  (d) $k$-NN  (e) SVM  (f) MLP

Figure 3: **Decision boundaries of Llama-3-8B with implicit reasoning where learning strategies are specified.** The figures show the decision boundaries of Llama-3-8B with prompts where machine learning strategies are specified. According to these figures, we can observe that the positions of decision boundaries do not change significantly except for becoming more fragmented at the edges.

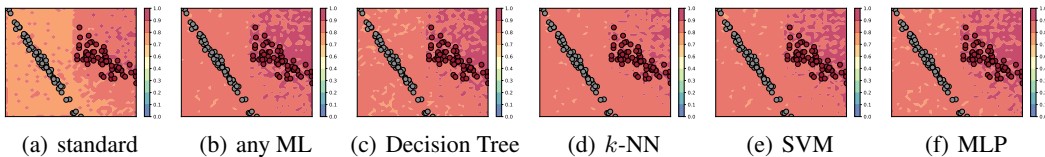

(a) standard  (b) any ML  (c) Decision Tree  (d) $k$-NN  (e) SVM  (f) MLP

Figure 4: **Confidence of predictions of Llama-3-8B with implicit reasoning where learning strategies are specified.** The figures show the "subjective" confidence of Llama-3-8B in its predictions of query data with the prompts where machine learning strategies are specified. According to the comparison between Fig. (a) and Figs. (b)-(f), we can observe that specifying learning strategies increases the confidence of Llama-3-8B in its predictions. Moreover, we can also observe that Llama-3-8B is consistently confident in its predictions of data in the upper right part of the plane.

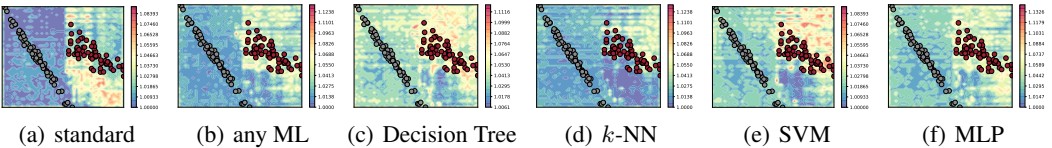

(a) standard  (b) any ML  (c) Decision Tree  (d) $k$-NN  (e) SVM  (f) MLP

Figure 5: **Perplexities of predictions of Llama-3-8B with implicit reasoning where learning strategies are specified.** The figures show the perplexities, which can be treated as the "objective" confidence, of Llama-3-8B in its predictions of query data with the prompts where machine learning strategies are specified. The low perplexity indicates high confidence. According to the figures, we can observe that specifying learning strategies increases the perplexities of Llama-3-8B in its predictions. Interestingly, the areas of high perplexity correspond to the high confidence in Fig. 4.

The decision boundaries are visualized in Fig. 3. According to Figs. 3(b)-3(f), we can observe that the positions of decision boundaries are not modified significantly compared to Fig. 3(a). However, due to the query data near the decision boundaries, the decision boundaries are slightly more fragmented. This can be further quantitatively demonstrated by smooth scores in Table 2. As shown in the table, we can observe that the smooth scores of decision boundaries derived from Llama-3-8B prompted by specified ML strategies are smaller than those from Llama-3-8B prompted by the standard prompt.

Moreover, we also probe the confidence of Llama-3-8B in its predictions. Specifically, we require Llama-3-8B to evaluate its confidence by itself (subjective confidence) and also calculate the perplexity of its responses (objective confidence). The results are visualized in Figs. 4 and 5, respectively. For the confidence, compared to Fig. 4(a), we can observe that Llama-3-8B becomes more confident in its predictions when learning strategies are specified in prompts (see Figs. 4(b)-4(f)). In addition, consistent among all cases, Llama-3-8B is more confident in the predictions of query data in the upper right region. Such a phenomenon implies some "preference" of the data cluster (i.e., Class 1).

For perplexity (Fig. 5), we observe that the perplexities of Llama-3-8B in its predictions also increase holistically, which indicates the confidence of Llama-3-8B in its predictions decreases. This indicates that specifying learning strategies for LLMs does not help in the reasoning. An interesting phenomenon here is that the areas with high perplexity match the areas with high confidence in Fig. 4. Specifically, the upper right region of the plane (Class 1), where Llama-3-8B "subjectively" prefers and is more confident, is actually the region where the LLM is unconfident in its calculations. Such a contradictory phenomenon implies different preferences of LLMs in the reasoning and calculation.

### 4.3 FURTHER EXPLORATION OF EXPLICIT REASONING

Although the decision boundary of explicit reasoning is more fragmented, two phenomena are worth noticing. On the one hand, LLMs tend to leverage machine learning (or data analysis) strategies to

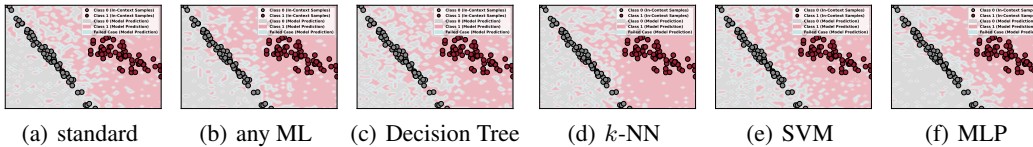

(a) standard    (b) any ML    (c) Decision Tree    (d) $k$-NN    (e) SVM    (f) MLP

Figure 6: **Decision boundaries of Llama-3-8B with explicit reasoning where learning strategies are specified.** The figures show the decision boundaries of Llama-3-8B in the way of explicit reasoning with prompts where ML strategies are specified. According to these figures, we can easily observe that the decision boundaries do not change significantly and are consistently fragmented.

Table 3: Smooth scores of decision boundaries of both conventional ML methods (scikit-learn) and explicit reasoning of LLMs prompted with specified machine learning methods (Specified ER).

| Case | Standard | Any ML | Decision Tree | $k$-NN | MLP | SVM |
|---|---|---|---|---|---|---|
| Baseline (sklearn) | - | - | 0.9868 | 0.9854 | 0.9854 | 0.9830 |
| Llama-3-8B (Specified ER) | 0.6809 | 0.6465 | 0.6465 | 0.6465 | 0.6465 | 0.6465 |

(a) standard    (b) SC-CoT-5    (c) SC-CoT-11    (d) standard    (e) SC-CoT-5    (f) SC-CoT-11

Figure 7: **Decision boundaries and perplexity of Llama-3-8B with self-consistency CoT (SC-CoT) where learning strategies are specified in prompts.** The figures show the decision boundaries and perplexity of Llama-3-8B in the way of CoT and SC-CoT reasoning with prompts where ML strategies are specified. According to these figures, we can observe that decision boundaries become even worse, though predictions in the right region are improved. Meanwhile, we also notice that the perplexities of Llama-3-8B in predictions decrease in the right part and increase in the left part.

address the classification tasks. On the other hand, the execution of the strategies (e.g., $k$-NN) is incorrect. Thus, in this section, we consider two aspects: (1) whether specifying learning strategies contributes to improving the reasoning of LLMs; and (2) whether voting predictions from sampling multiple reasoning trajectories facilitates alleviating the incorrect executions of learning strategies.

In order to answer the first question, we follow Section 4.2 to have LLMs perform classification tasks with prompts, where learning strategies are specified. The decision boundaries are visualized in Fig. 6 and quantitatively measured in Table 3. According to Figs. 6(b)-6(f), we can observe that the decision boundaries remain fragmented. Meanwhile, according to the table, we can observe that the smooth score of the cases, where learning strategies are specified, is smaller than that of the standard case. These quantitative results indicate that decision boundaries become more fragmented when learning strategies are specified in prompts. This phenomenon is consistent with that in the case of implicit reasoning and indicates that specifying learning strategies does not help improve the reasoning.

To answer the second question, we further apply the self-consistency CoT (SC-CoT, (Wang et al., 2023)) in the explicit reasoning without specifying any learning strategies in prompts. The decision boundaries and the corresponding perplexities are visualized in Fig. 7. According to Figs. 7(a)-7(c), we can observe that the decision boundaries are still fragmented. However, we notice that the predictions of query data in the right region are improved. Meanwhile, we can also observe interesting phenomena in the perplexities of Llama-3-8B (Figs. 7(d)-7(f)). Specifically, when SC-CoT is applied, the perplexity of predictions in the right region decreases, which implies that Llama-3-8B becomes more confident in its predictions. In contrast, the perplexity of predictions in the left region increases, which implies that Llama-3-8B becomes less confident in its predictions. These phenomena, to some extent, correspond to the phenomenon that the predictions regarding queries close to Class 1 (right) are improved, while the predictions regarding queries close to Class 0 (left) become even worse.

## 5 A DEEPER EXPLORATION OF BEHAVIORS OF LLMS ON CLASSIFICATION

Based on both quantitative and qualitative results in Section 4, we can summarize that LLMs can achieve better prediction results in classification tasks in the way of implicit reasoning without any specific instructions. This phenomenon implies that LLMs intrinsically possess a basic capability

Table 4: **The KL-divergence results of predictions on the linear classification task, respectively, between Llama-3-8B and itself, conventional machine learning algorithms and themselves, and Llama-3-8B and conventional machine learning algorithms.** $\bar{d}_{\text{LLM}}$ denotes the KL-divergence of predictions between Llama-3-8B and itself, $\bar{d}_{\text{Conv}}$ denotes the KL-divergence of predictions of conventional machine learning and themselves, and $\bar{d}_{\text{LLM}\to\text{method}}$ denotes the KL-divergence of predictions between Llama-3-8B and conventional ML methods, where $\text{method} \in \{\text{DT}, \text{kNN}, \text{MLP}, \text{SVM}\}$.

| Case | $\bar{d}_{\text{LLM}}$ | $\bar{d}_{\text{Conv}}$ | $\bar{d}_{\text{LLM}\to\text{DT}}$ | $\bar{d}_{\text{LLM}\to\text{kNN}}$ | $\bar{d}_{\text{LLM}\to\text{MLP}}$ | $\bar{d}_{\text{LLM}\to\text{SVM}}$ |
|---|---|---|---|---|---|---|
| Standard (LLM) | 6.76 | - | 7.03 | 7.00 | 6.99 | 6.99 |
| Decision Tree (LLM) | 6.54 | 7.54 | 6.83 | 6.80 | 6.79 | 6.78 |
| $k$-NN (LLM) | 6.70 | 7.72 | 6.97 | 6.93 | 6.92 | 6.92 |
| MLP (LLM) | 6.57 | 7.55 | 6.84 | 6.80 | 6.80 | 6.79 |
| SVM (LLM) | 6.67 | 7.53 | 6.93 | 6.90 | 6.89 | 6.88 |

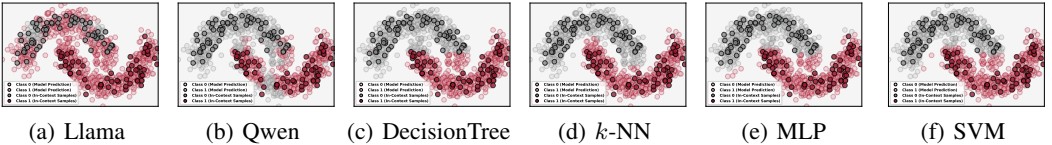

  (a) Llama      (b) Qwen      (c) DecisionTree      (d) $k$-NN      (e) MLP      (f) SVM

Figure 8: **Visualizations of predictions of Llama-3-8B, Qwen-2.5-7B, and conventional machine learning methods on the in-distribution query data in the linear classification task**. Fig. (a)&(b) visualize the predictions of Llama-3-8B and Qwen-2.5-7B, respectively. Figs. (c)-(f) visualize the predictions of conventional ML methods. According to the visualization results, we can observe that the two LLMs show similar behavior to the conventional machine learning algorithms. Such a phenomenon is consistent with the quantitative results in Table 4, where the KL divergence of the predictions between Llama-3-8B (or Qwen-2.5-7b) and conventional ML algorithms are similar.

  (a) Llama      (b) Qwen      (c) DecisionTree      (d) $k$-NN      (e) MLP      (f) SVM

Figure 9: **Visualizations of predictions of Llama-3-8B, Qwen-2.5-7B, and conventional ML methods on the in-distribution query data in the moon classification task**. Fig. (a)&(b) visualize the predictions of Llama-3-8B and Qwen2.5-7B, respectively. Figs. (c)-(f) visualize the predictions of conventional ML methods. According to the visualizations, we can observe that Llama-3-8B fails to perform the moon classification, while Qwen2.5-7B reveals similar behavior to conventional ML algorithms. Both phenomena match the quantitative results respectively reported in Tables 8 and 14.

of handling classification tasks. However, due to the black-box nature of implicit reasoning, such a capability remains unclear. Thus, in this section, we propose to probe the underlying capability of LLMs in implicit reasoning from the angle of distribution. Complete discussions are in Appendix F.

## 5.1 WHETHER LLMs IMPLICITLY LEVERAGE MACHINE LEARNING ALGORITHMS?

In our study of the implicit reasoning of LLMs in Fig. 3(a), we can observe that the decision boundary derived from implicit reasoning with the standard prompt resembles that derived from conventional machine learning algorithms (e.g., $k$-NN (Fig. 1(b)), SVM (Fig. 1(c)), and MLP(Fig. 1(d)). This phenomenon raises a question: *Do LLMs perform classification by implicitly leveraging ML methods?*

Consider two data distributions in a space. For arbitrary two sets of data clusters sampled from the two distributions in the space, there exists a decision boundary, and the distribution, where the decision boundary lies, is closely related to the given classification strategies (e.g., ML algorithms/models). For example, the decision boundaries derived from ten thousand times running of the $k$-NN algorithm with different training and evaluation data sampled from the space will converge to a specific distribution. Such a distribution can be considered to describe the characteristics of the given classification strategy.

**Definition 5.1.** *Given a set of data distributions in a space, and two classification strategies, where one is the test strategy and the other is the reference strategy, two sets of decision boundaries can be derived by running the two classification strategies, respectively, with different sets of in-context and query data randomly sampled from the space. Then, the average KL-divergence $\bar{d}_{\text{ref}}$ regarding arbitrary two decision boundaries derived from the reference strategy and the average KL-divergence*

$\bar{d}_{\text{test}\to\text{ref}}$ *regarding arbitrary two decision boundaries respectively derived from the test and the reference strategies can then be calculated. In this way, the behaviors of the two classification strategies are the same if the gap between the two KL-divergence scores, $d_{\text{ref}}$ and $d_{\text{test}\to\text{ref}}$, is small.*

Note that, in Definition 5.1, we primarily focus on the difference between the two KL-divergence scores, rather than the magnitude of a single KL-divergence score, since the KL-divergence of decision boundaries may be large due to the sampling of the in-context and query data. The key here is the difference between the two KL-divergence scores. Specifically, if LLMs perform classification in the same way as conventional machine learning methods, the $\bar{d}_{\text{ref}}$ and $\bar{d}_{\text{test}\to\text{ref}}$ will be the same.

To validate the definition, we first generate two 2D data distributions and then randomly sample 10 sets of in-context and query data. Then, we generate two sets of decision boudaries from Llama-3-8B and conventional machine learning methods, respectively, with the 10 sampled data sets, and measure the two KL-divergence scores $\bar{d}_{\text{ref}}$ and $\bar{d}_{\text{test}\to\text{ref}}$. Llama-3-8B is prompted with both the standard prompt and the prompts in which learning strategies are specified. The results are reported in Table 4. According to the table, we can observe that, in the case that Llama-3-8B is prompted with the standard prompt, the difference between $\bar{d}_{\text{Conv}}$ and $\bar{d}_{\text{LLM}\to\text{method}}$ is small ($\sim 0.5$). However, when Llama-3-8B is prompted with prompts where learning strategies are specified, the differences between $\bar{d}_{\text{Conv}}$ and $\bar{d}_{\text{LLM}\to\text{method}}$ are relatively larger. The results are consistent with the phenomenon that specifying learning strategies results in the deterioration of decision boundaries in previous sections.

Thus, with the observations of the small difference between the two KL-divergence scores, we can summarize that the behavior of Llama-3-8B on the linear classification task with implicit reasoning in ICL is consistent with that of conventional machine learning algorithms conducted by scikit-learn.

## 5.2 Are Decision Boundaries of LLMs Affected by Distribution of Query Data?

Typically, the query data are generated by splitting the plane, where the in-context data lie, uniformly into a set of grids. Actually, in this case, the query data include both in-distribution (ID) data and out-of-distribution (OOD) data. Specifically, the data near the underlying distribution of the in-context data can be treated as in-distribution data, while the others are out-of-distribution data. This further inspires us to consider whether the distribution of query data affects the decision boundaries of LLMs.

To solve this concern, we follow the setting in the previous section, and evaluate LLMs (Llama-3-8B and Qwen-2.5-7B) on both linear and moon classification tasks, where in-context and query data are sampled from the same distribution. The prediction results are visualized in Figs. 8 and 9. For simple classification tasks, such as the linear classification task, both Llama-3-8B and Qwen-2.5-7B achieve similar prediction results to conventional machine learning algorithms. The quantitative results of both Llama-3-8B and Qwen-2.5-7B (Tables 4 and 12) reveal that the differences between $\bar{d}_{\text{Conv}}$ and $\bar{d}_{\text{LLM}\to\text{method}}$ are consistently small. These phenomena indicate that LLMs, in fact, can achieve good performance in classification tasks where the in-context and query data are in-distribution data.

However, for the complex classification tasks (e.g., the moon classification task), the two LLMs perform completely different. According to the visualization results in Fig. 9(a), we can observe that Llama-3-8B fails to correctly predict the labels of query data. Specifically, Llama-3-8B assigns most query data that belongs to Class 0 to Class 1. According to the quantitative results in Table 8, there is an evident gap between $\bar{d}_{\text{Conv}}$ and $\bar{d}_{\text{LLM}\to\text{method}}$. In contrast, for Qwen-2.5-7B (Fig. 9(b)), the predictions are basically consistent with those derived from conventional machine learning algorithms. This phenomenon is also consistent with the quantitative results in Table 14, where the gap is small.

Overall, LLMs can achieve good performance similar to conventional machine learning algorithms on those machine learning classification tasks, where both in-context and query data are sampled from the same distribution. However, such a capability is pre-defined by the pretraining of the LLMs.

## 6 Conclusion

In this paper, we take a further step in exploring the in-context capability of LLMs in ML classification tasks. Specifically, we first probe the behaviors of LLMs with various implicit and explicit reasoning paradigms qualitatively and quantitatively. To further probe the underlying capability of implicit reasoning, which achieves better boundaries, we propose a new method to measure the difference of the KL-divergence regarding the decision boundaries, respectively derived from reference and test strategies. In this way, we determine that the behavior of implicit reasoning is consistent with the behaviors of conventional ML algorithms. However, such a capability is limited by the pretraining.

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

APPENDIX

## BROADER IMPACT

In this paper, we mainly focus on exploring the in-context capability of LLMs on typical machine learning classification tasks. The main goal of our paper is to determine the behaviors of LLMs in classification tasks with in-context learning. Since the LLM community mainly treats LLMs as black-box models and leverages them for content generation. There exist risks that harmful information is generated with elaborately designed prompts, such as jailbreaking the LLMs for harmful information with elaborately selected demonstrations in the way of in-context learning. From this perspective, works like our paper, which explores the underlying mechanisms of in-context learning, may help researchers achieve a more comprehensive understanding of the behaviors of LLMs in in-context learning and, in turn, develop appropriate strategies to avoid the undesirable cases. In this way, we can ensure that the developments of artificial intelligence techniques are constrained in a safe scope.

## LIMITATIONS

In this paper, we mainly take a further step in exploring the in-context learning capability of LLMs. By probing the behaviors of LLMs with various kinds of reasoning paradigms (e.g., implicit and explicit reasoning), we achieve a comprehensive understanding of the behaviors of LLMs in typical ML classification tasks. In order to further explore the underlying mechanism of the implicit reasoning in classification tasks, we then leverage statistical tools to probe the connection between the behavior modes of LLMs and conventional machine learning algorithms conducted by the scikit-learn tools.

One potential limitation of our work is that the explorations of the behaviors of LLMs are not comprehensive enough due to the lack of tools. Currently, as far as we know, there are few works focusing on developing tools for exploring the underlying mechanisms of LLMs. In fact, developing such tools is essential since the research works, which are based on the black-box nature of LLMs, are not solid enough to provide users and researchers with valuable insights. In this paper, in order to alleviate such a limitation, we propose some basic and simple statistical tools, such as smooth score and differences between KL-divergence. Even so, our work remains constrained by the limitation.

# A  RELATED WORK

In-context learning is an important emergent capability of large language models derived from the pretraining on huge amounts of corpora. Abundant works have been done to explore this capability.

**In-context Learning.** With the development of model architectures in deep learning (e.g., Transformer (Vaswani et al., 2017)), the sizes of models have been significantly scaled (Brown, 2020; Achiam et al., 2023; Chowdhery et al., 2023). Along with the increased scale of models, the capability of these foundation models is also evidently improved. One of these impressive abilities is in-context learning, which enables LLMs to perform previously unseen tasks without extra training steps. The key idea of in-context learning is learning to perform tasks with only a few samples, which are treated as demonstrations. This resembles the decision-making process of human beings, where the common features are extracted from the demonstrations and applied to the analogical new tasks (Winston, 1980). Currently, in-context learning is mainly performed via prompts (Liu et al., 2023). Specifically, by elaborately designing the instructions and the demonstrations of the tasks, LLMs can follow these contents and mimic the behavior to complete complex tasks, such as reasoning (Wei et al., 2022). However, why in-context learning achieves such impressive performance remains an open problem.

Some previous works try to build a connection between in-context learning and gradient-based meta-learning (Finn et al., 2017; Finn and Levine, 2017; Gordon et al., 2018; Lee et al., 2019; Zhou et al., 2019; Rajeswaran et al., 2019). For example, Von Oswald et al. (2023) demonstrates that training Transformers on auto-regressive objectives is closely related to gradient-based meta-learning. Meanwhile, Dai et al. (2023) demonstrates that the calculation of attention can be treated as a dual form of gradient descent. In such a case, the transformer models can thus be viewed as meta-optimizers. In addition, in-context learning can also be explained from the perspective of gradient descent. For instance, Ahn et al. (2024) observe that the transformer performs preconditioned gradient descent when the parameters are trained to converge. Zhang et al. (2024) demonstrates that the transformer is able to achieve a competitive prediction error with the best linear prediction on a new prediction task. In addition to these theoretical works, some other works also try to explore in-context learning from a practical perspective. For example, Wei et al. (2023) studies large language models with respect to the size of models. In this work, prior knowledge, such as labels, is demonstrated to be essential to the performance of in-context learning. Lampinen et al. (2022) demonstrates that plugging explanations in in-context samples can significantly improve the performance of LLMs.

**Discriminative Tasks with LLMs.** LLMs have been demonstrated to be powerful on content generation tasks, such as reasoning (Wei et al., 2022) and Q&A (Achiam et al., 2023). However, such a capability on discriminative tasks, such as machine learning classification tasks, has not been well explored. Recently, Shi et al. (2023) looks into the discriminative capability of LLMs by transferring the discriminative tasks into language descriptions and demonstrates that the performance of LLMs is closely related to the contents in prompts. Specifically, if irrelevant information is contained in the contents, the performance will be significantly damaged. Besides, Xiao et al. (2024) proposes to perform discriminative tasks by optimizing the LLM with LLMs. Specifically, in this work, two LLMs are adopted respectively as the learner and the optimizer. The prompts, which are used in the learner, are treated as some kind of "parameters" and are optimized by the optimizer LLM with specific hyperparameters, such as learning rate. The results show that the performance on discriminative tasks can be improved after a few learning steps. In order to examine the discriminative capability of LLMs, Zhao et al. (2024) proposes to make LLMs perform conventional classification tasks and probe the decision boundaries. In this work, LLMs are found to be irregularly incompetent and fail to achieve smooth decision boundaries if the model is not fine-tuned in an appropriate way. Our work is inspired by Zhao et al. (2024). In this paper, we propose to dive into the behavior of LLMs in discriminative tasks to figure out the reasons for the failure in the simple classification tasks.

# B    DETAILED TASK SETTINGS

In this section, we provide more details regarding the machine learning classification tasks adopted in our work. In this paper, we propose to study the in-context learning capability of large language models on simple machine learning classification tasks. To this end, we mainly consider three types of classification tasks: conventional 2D classification tasks, 3D classification tasks (a special case of high-dimensional classification tasks), and classification tasks where in-context and query data are sampled from the same distribution. In the following sections, we introduce the three types of tasks.

## B.1    CONVENTIONAL 2D CLASSIFICATION TASKS

In this section, we provide details for the generation of the conventional 2D classification tasks adopted in our paper. Specifically, the 2D classification tasks adopted in this paper include linear classification, circle classification, and moon classification. In the main paper, we mainly consider linear classification tasks. The hyperparameter settings in paper are consistent with Zhao et al. (2024).

Following Zhao et al. (2024), all three types of 2D classification tasks adopted in this paper are generated with the existing functions: make_classification, make_circles, and make_moons in scikit-learn tools (Pedregosa et al., 2011). In this paper, for simplicity, we mainly consider binary classification tasks. Specifically, in the linear classification task, linearly separable data are generated around a hypercube. In the circle classification, two circles of data, where the smaller circle lies in the larger one, are generated. In moon classification, two interleaving half-circles are generated.

By default, in each class, the number of samples in each class is set to 64. The class_sep parameter in the linear classification task generation function is randomly sampled from the range $[1.5, 2.0]$; the factor parameter in the circle classification task generation function is randomly sampled from $[0.1, 0.4]$; and the noise parameter in the moon classification task generation function is randomly sampled from $[0.05, 0.1]$. Moreover, in circle classification tasks, the parameter noise is set to 0.03.

In the generation of each task, we first generate 2000 data samples, where each class includes 1000 data samples. Then, we randomly selected 64 data samples for each class, respectively, as the in-context data of the task. For the reproducibility of the experiments, we use the random seed 11.

## B.2    3D CLASSIFICATION TASKS

In our paper, we also study the decision boundaries of LLMs in high-dimensional classification tasks. However, a problem of studying high-dimensional classification tasks is the amount of query data. As we mentioned in Section 2, the query data in our classification tasks are generated by splitting the plane, where the in-context data lie, uniformly into $N^d$ grids, where $N$ denotes the number of grids for each dimension, and $d$ denotes the number of dimensions. In this case, as the number of dimensions increases, the amount of query data also increases drastically. For example, if $N = 20$, the number of query data is 2500 in the 2D classification task, while the number of query data in the 3D classification task is 12500. In this case, the consumption of each experiment will also increase.

To avoid this problem, for high-dimensional classification tasks, we propose to sample 2500 query data from the $N^d$ candidate data samples. However, a drawback of this strategy is that the sampled data points are quite sparse in their original set. Thus, with both aspects taken into consideration, we make a trade-off in our paper. We propose to probe the behaviors of LLMs on high-dimensional classification tasks by selecting the 3D classification task as the representative of high-dimensional classification tasks. On the one hand, the 3D classification task is a special case of high-dimensional classification tasks. On the other hand, the sampled query data are not sparse in the original space.

Concretely, in the generation of a 3D classification task, we first follow the 2D task to sample 2000 data samples randomly and then randomly select 128 in-context data from these data samples. In this process, we use the random seed 11. With the sampled in-context data, we then split the space, where the in-context data lie, uniformly into $N^3$ grids (in our paper, $N = 50$). Then, with the random seed 42, we further select 2500 query data points from the 12500 candidate query data samples above.

### B.3    IN-DISTRIBUTION TASKS

In our paper, we find that the query data adopted in the previous work (Zhao et al., 2024) actually include both in-distribution and out-of-distribution data. In conventional learning tasks, out-of-distribution data usually result in poor generalization performance. In order to figure out whether the decision boundaries of LLMs are affected by the distribution discrepancies between the in-context and query data, we propose to further probe the behaviors of LLMs on classification tasks, where both in-context and query data are sampled from the same distribution in a given space. Meanwhile, such a classification task is also adopted to explore the underlying behavior of implicit reasoning.

Specifically, in the in-distribution tasks, we first generate 10000 data samples for each class with the random seed 11. For the experiment of probing the underlying behavior of implicit reasoning, we randomly sample 10 sets of data to perform the classification task. For each set, each class owns 64 in-context and 500 query data samples. The 10 sets of data are sampled with the random seeds 40-49. For the visualization of predictions of LLMs, we use the data set sampled with the random seed 42.

### B.4    COMPUTATIONAL RESOURE INFORMATION

In this paper, most of our experiments are conducted on a single NVIDIA GeForce RTX 3090 GPU (24GB), while the remaining cases are conducted on a single NVIDIA RTX A6000 GPU (48 GB).

## C  DECISION BOUNDARIES OF IMPLICIT & EXPLICIT REASONING OF MISTRAL AND QWEN

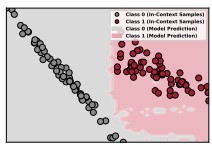 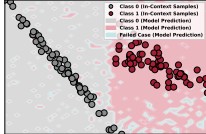 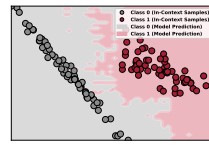 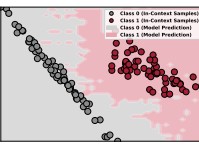

(a) Mistral-8B (IR))    (b) Mistral-8B (ER)    (c) Qwen-2.5-7B (IR)    (d) Qwen-2.5-7B (ER)

Figure 10: **Visualization results of decision boundaries on Mistral-8B and Qwen-2.5-7B.** We evaluate more representative LLMs, Mistral-8B and Qwen-2.5-7B, on the linear classification tasks in order to validate the generality of the observations mentioned in Section 3.1. According to the presented figures, the observations of decision boundaries on these two LLMs are consistent with those on Llama-3-8B. Specifically, both decision boundaries of implicit and explicit reasoning are fragmented, and the decision boundary of implicit reasoning is better than that of explicit reasoning.

In Section 3.1, we propose to obtain a comprehensive understanding of the behaviors of large language models in the linear classification task in the way of in-context learning. For simplicity, we only provide the visualization results of Llama-3-8B in the main paper. Here, in order to validate the generality of the aforementioned observations, we further evaluate more LLMs on the linear classification task. Specifically, in this section, we evaluate Mistral-8B (Jiang et al., 2023) and Qwen-2.5-7B (Yang et al., 2024a) on the linear classification task to explore the ICL capabilities.

The decision boundaries of the two LLMs are visualized in Fig. 10. According to the visualization results, we can observe that the two observations remain in Mistral-8B and Qwen-2.5-7B. Specifically, the decision boundaries derived from LLMs are fragmented, and the decision boundary obtained from implicit reasoning is better than that obtained from explicit reasoning. Thus, we can demonstrate that the two observations (Observation 1 & 2) in Section 3.1 widely exist among large language models.

# D  MORE DECISION BOUNDARIES OF LLMS WITH IMPLICIT & EXPLICIT REASONING

In this paper, we are motivated by the decision boundaries derived from LLM, respectively, with implicit and explicit reasoning paradigms. According to our visualization results in Fig. 1, two main phenomena are observed. Firstly, the decision boundaries derived from both implicit and explicit reasoning are fragmented. This phenomenon is consistent with the observation in Zhao et al. (2024). Moreover, we also observe that the decision boundary derived from implicit boundary is better than that derived from explicit reasoning. However, whether such phenomena are general among LLMs and classification tasks remains unclear. To figure out this problem, in this section, we propose to evaluate more LLMs on more classification tasks. Specifically, for LLMs, we further take Mistral-8B (Jiang et al., 2023) and Qwen-2.5-7B (Yang et al., 2024a) into consideration for a comprehensive study. Moreover, for classification tasks, we adopt circle and moon classification.

## D.1  CONFIDENCE AND PERPLEXITY OF EXPLICIT REASONING OF LLAMA-3-8B

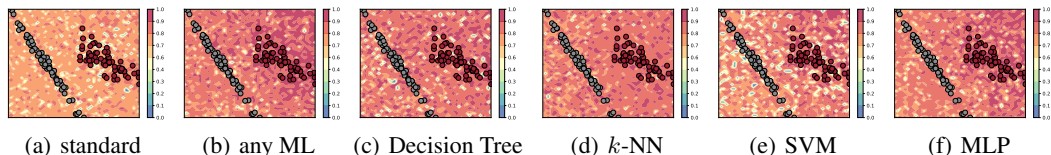

| (a) standard | (b) any ML | (c) Decision Tree | (d) $k$-NN | (e) SVM | (f) MLP |

Figure 11: **Confidence in predictions of Llama-3-8B with explicit reasoning where learning strategies are specified.** The figures show the confidence of Llama-3-8B in its predictions of query data in the way of explicit reasoning with the prompts where machine learning strategies are specified.

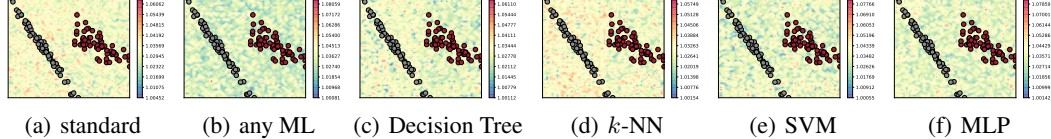

| (a) standard | (b) any ML | (c) Decision Tree | (d) $k$-NN | (e) SVM | (f) MLP |

Figure 12: **Perplexities in predictions of Llama-3-8B in the linear classification task with explicit reasoning.** The figures show the perplexities of Llama-3-8B in its predictions in the linear classification task in the way of explicit reasoning with both the standard prompt and prompts where machine learning strategies are specified. The low perplexity indicates high confidence of LLMs.

In Section 4.3, we explore whether specifying machine learning strategies facilitates improving the decision boundaries of explicit reasoning of LLMs and, in turn, improving the smoothness of the boundaries. The decision boundaries are visualized in Fig. 6 and quantitatively measured in Table 3.

In order to further figure out the behavior of Llama-3-8B in explicit reasoning. We follow the case of implicit reasoning to examine the confidence and perplexity of Llama-3-8B in explicit reasoning. The visualization results of confidence and perplexity are, respectively, reported in Fig. 11 and Fig. 12.

For confidence, there are two phenomena worth noticing. On the one hand, the explicit reasoning also reveals the "preference", which is observed in implicit reasoning, for the data in the upper right region (Class 1) (Fig. 11(a)). Meanwhile, when learning strategies are specified in prompts, the "preference" is also enhanced (see Figs. 11(b)-11(f)). Llama-3-8B still reveals high confidence for query data that are close to Class 1. On the other hand, compared to explicit reasoning with the standard prompt, the confidence in predictions with prompts, where learning strategies are specified, also increases. Such a phenomenon is also consistent with the case of implicit reasoning with specified learning strategies.

For perplexity, we can observe that the perplexity for all cases of explicit reasoning, regardless of whether with the standard prompt or with the prompts where learning strategies are specified, is relatively higher than that of implicit reasoning cases. However, compared to the standard case of explicit reasoning (Fig. 12(a)), the perplexity in predictions in the cases, where learning strategies are specified, decreases slightly, such as in the cases of any ML (Fig. 12(b)), Decision Tree (Fig. 12(c)), SVM (Fig. 12(e)), and MLP (Fig. 12(f)). Such phenomena are different from implicit reasoning. However, the improved perplexity does not further contribute to improving the decision boundaries.

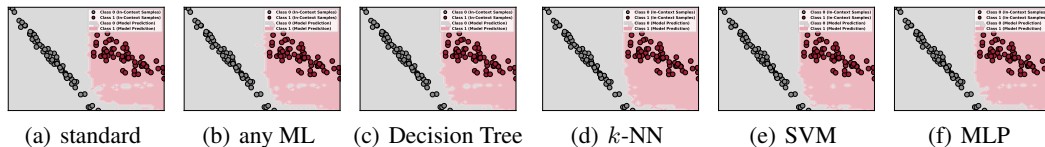

| (a) standard | (b) any ML | (c) Decision Tree | (d) $k$-NN | (e) SVM | (f) MLP |

Figure 13: **Decision boundaries of Mistral-8B with implicit reasoning where learning strategies are specified.** The figures show the decision boundaries of Mistral-8B with prompts where machine learning strategies are specified. According to these figures, we can observe that the positions of decision boundaries do not change significantly except for becoming more fragmented at the edges.

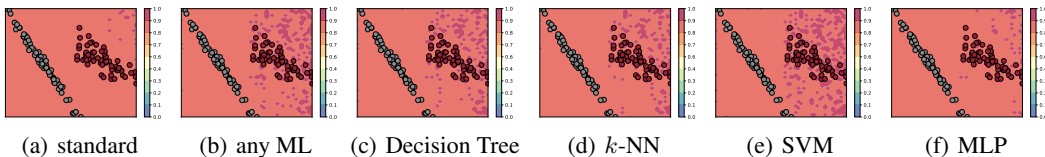

| (a) standard | (b) any ML | (c) Decision Tree | (d) $k$-NN | (e) SVM | (f) MLP |

Figure 14: **Confidence of predictions of Mistral-8B with implicit reasoning where learning strategies are specified.** The figures show confidence of Mistral-8B in its predictions of query data in the way of implicit reasoning with the prompts where machine learning strategies are specified.

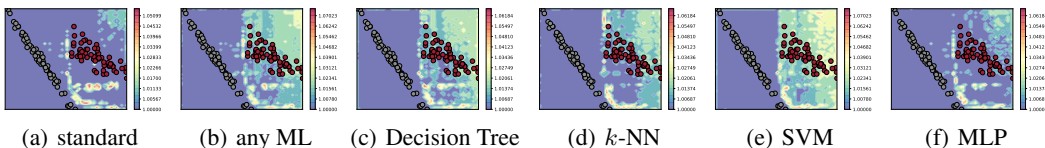

| (a) standard | (b) any ML | (c) Decision Tree | (d) $k$-NN | (e) SVM | (f) MLP |

Figure 15: **Perplexities in predictions of Mistral-8B in the linear classification task with implicit reasoning.** The figures show the perplexities of Mistral-8B in its predictions in the linear classification task in the way of implicit reasoning with both the standard prompt and the prompts, where machine learning strategies are specified. The low perplexity indicates high confidence of LLMs.

Overall, with all these observations, we can summarize that, for Llama-3-8B, extra specific instructions (e.g., specified learning strategies) will increase its subjective confidence in prediction. From the perspective of decision boundaries, such confidence, to some extent, is more like a kind of hallucination. Moreover, compared to the standard case of implicit reasoning, both specific instructions and explicit reasoning increase the perplexity, which implies that LLMs possess the intrinsic capability of handling classification tasks, while excessive reasoning steps and instructions damage the capability.

### D.2 DECISION BOUNDARIES OF IMPLICIT REASONING ON MORE LLMS AND CLASSIFICATION TASKS

In previous sections, we mainly explore the in-context learning capability of LLMs on the linear classification task from the perspective of Llama-3-8B. In this section, in order to further explore the behaviors of LLMs on more classification tasks with the in-context learning paradigm, we propose to examine the behavior of other LLMs, such as Mistral-8B (Jiang et al., 2023) and Qwen-2.5-7B (Yang et al., 2024a), on other classification tasks, such as the circle classification task and the moon classification task. In this section, we mainly take implicit reasoning into consideration.

### D.2.1 MISTRAL-8B ON LINEAR CLASSIFICATION TASK WITH IR

The visualization results of decision boundaries, confidence, and perplexity of Mistral-8B on the linear classification task with implicit reasoning are reported in Figs. 13, 14, and 15, respectively. According to the figures, we can observe the following several phenomena. (1) For decision boundaries, the implicit reasoning consistently achieves relatively better decision boundaries on the linear classification task than the cases where learning strategies are specified, and decision boundaries in both the standard case and the cases where learning strategies are specified are similar. (2) For confidence, Mistral-8B is more confident in its predictions than Llama-3-8B. With learning strategies specified in prompts, we can observe that the confidence in the predictions in the right region (Class 1) increases. Moreover, we can observe that Mistral-8B also reveals a preference for Class 1 (the upper right region). (3) For the perplexity in the calculation, we can observe that the perplexity of Mistral-8B in the standard case is relatively lower than that of Llama-3-8B, especially in the regions near the in-context data. However, when learning strategies are specified in prompts, the perplexity of

Mistral-8B in the right region increases significantly, while the perplexity in the left region remains low. (4) Consistent with that in Llama-3-8B, the region where Mistral-8B is more confident (the upper right region) corresponds to the region, where the perplexity is high. With all these aspects taken into consideration, we can observe that the behaviors of Mistral-8B resembles those of Llama-3-8B.

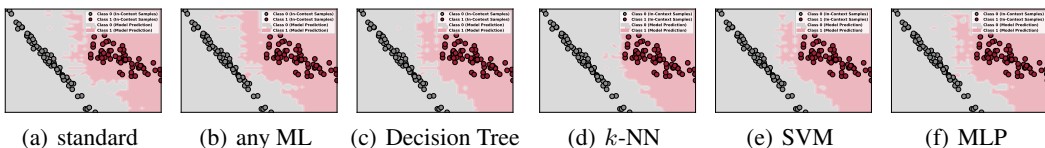

| (a) standard | (b) any ML | (c) Decision Tree | (d) $k$-NN | (e) SVM | (f) MLP |

Figure 16: **Decision boundaries of Qwen-2.5-7B with implicit reasoning where learning strategies are specified.** The figures show the decision boundaries of Qwen-2.5-7B with prompts where machine learning strategies are specified. According to these figures, we can observe that the positions of decision boundaries change slightly and the boundaries are improved slightly in some cases.

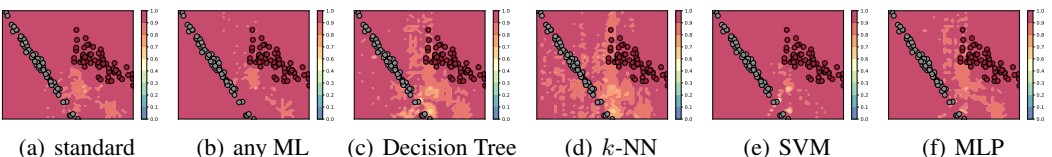

| (a) standard | (b) any ML | (c) Decision Tree | (d) $k$-NN | (e) SVM | (f) MLP |

Figure 17: **Confidence of predictions of Qwen-2.5-7B with implicit reasoning where learning strategies are specified.** The figures show confidence of Qwen-2.5-78B in its predictions of query data in the way of implicit reasoning with the prompts where machine learning strategies are specified.

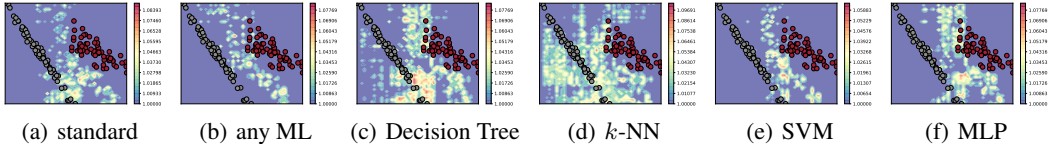

| (a) standard | (b) any ML | (c) Decision Tree | (d) $k$-NN | (e) SVM | (f) MLP |

Figure 18: **Perplexities in predictions of Qwen-2.5-7B in the linear classification task with implicit reasoning.** The figures show the perplexities of Qwen-2.5-7B in its predictions in the linear classification task in the way of implicit reasoning with both the standard prompt and prompts where machine learning strategies are specified. The low perplexity indicates high confidence of LLMs.

### D.2.2 QWEN-2.5-7B ON LINEAR CLASSIFICATION TASK WITH IR

The visualization results of decision boundaries, confidence, and perplexity of Qwen-2.5-7B on the linear classification task with implicit reasoning are reported in Figs. 16, 17, and 18, respectively. According to the visualization results, we can observe the following phenomena. (1) Consistent with Llama-3-8B and Mistral-8B, the decision boundaries derived from Qwen-2.5-7B are also fragmented in all cases. However, different from the two LLMs, when learning strategies are specified, the decision boundaries are improved slightly, such as in the case where the $k$-NN algorithm is specified. (2) For the confidence in prediction, we can observe that Qwen-2.5-7B is quite confident in its predictions, although the generated decision boundaries look worse than Llama-3-8B. Moreover, we also notice that the confidence changes in a random way when learning strategies are specified in prompts. Specifically, when learning strategies, such as Decision Tree (Fig. 17(c)) and $k$-NN (Fig. 17(d)), are specified in prompts, the confidence decreases significantly. However, when Qwen-2.5-7B is allowed to leverage any machine learning methods (Fig. 17(b)), the confidence increases slightly. (3) For the perplexity in prediction, similar to the confidence results, Qwen-2.5-7B reacts differently towards different learning strategies. Specifically, we can observe that the perplexity increases when Decision Tree and $k$-NN are specified in prompts. However, when arbitrary machine learning strategies are allowed in reasoning, the perplexity obviously decreases. Overall, we can observe that specifying learning strategies in prompts increases the perplexity of in the prediction.

### D.2.3 DECISION BOUNDARIES ON NON-LINEAR CLASSIFICATION TASKS WITH IR

In this section, we further explore the in-context learning capability of LLMs (Llama-3-8B, Mistral-8B, and Qwen-2.5-7B) on non-linear classification tasks (i.e., circle classification task and moon

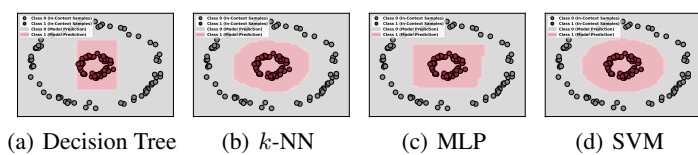

(a) Decision Tree     (b) $k$-NN     (c) MLP     (d) SVM

Figure 19: **Decision boundaries of conventional ML methods on the circle classification task.**

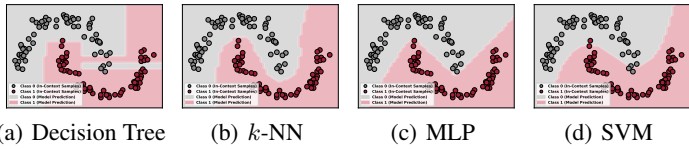

(a) Decision Tree     (b) $k$-NN     (c) MLP     (d) SVM

Figure 20: **Decision boundaries of conventional ML methods on the moon classification task.**

classification task). The visualization results of decision boundaries, confidence, and perplexity of all LLMs with implicit reasoning prompted by different prompts are respectively reported in Figs. 21-38.

First of all, we evaluate the performance of conventional machine learning methods on the two non-linear classification tasks. According to the visualizations in Figs. 19 and 20, we can observe that conventional machine learning methods can consistently obtain clear and smooth decision boundaries in the circle classification task. However, in the moon classification task, we can observe that Decision Tree fails to achieve a smooth decision boundary compared to the other machine learning methods.

Then, we examine the performance of the three LLMs. From the perspective of decision boundaries, in the circle classification task, except for Llama-3-8B, which fails to achieve smooth decision boundaries, both Mistral-8B and Qwen-2.5-7B can achieve relatively clearer decision boundaries, although they also fail to generate smooth decision boundaries (see Figs. 21-23). Moreover, in the moon classification task, we can observe that all LLMs fail to achieve clear decision boundaries. However, compared to Llama-3-8B, the decision boundaries obtained from Mistral-8B and Qwen-2.5-7B are relatively better. Meanwhile, for all LLMs, it is easy to find that specifying learning strategies in prompts does not facilitate improving the smoothness of the decision boundaries (see Figs. 30-32).

For the confidence, the phenomena in non-linear classification tasks are consistent with those observed in the linear classification task. Specifically, for Llama-3-8B and Mistral-8B, specifying learning strategies increases the confidence in their prediction. Both LLMs reveal a preference for Class 1. However, for Qwen-2.5-7B, the LLM reacts differently to different learning strategies. This phenomenon exists in both circle (see Figs. 24-26) and moon classification tasks (see Figs. 33-35).

For perplexity, in the circle classification task, we can observe that Llama-3-8B shows lower perplexity in the left region areas and higher perplexity in the right region areas in the standard case, while showing lower perplexity in the right region areas and higher perplexity in the left region areas (see Fig. 27). For Mistral-8B, we can observe that the perplexity increases, especially for the areas between the two classes (see Fig. 28). For Qwen-2.5-7B, we can observe that the LLM reacts differently to different learning strategies (see Fig. 29). In the moon classification task, for both Llama-3-8B and Mistral-8B, we observe that the perplexity increases when learning strategies are specified (see Figs. 36-37), while Qwen-2.5-7B reacts differently towards different learning strategies (see Fig. 38).

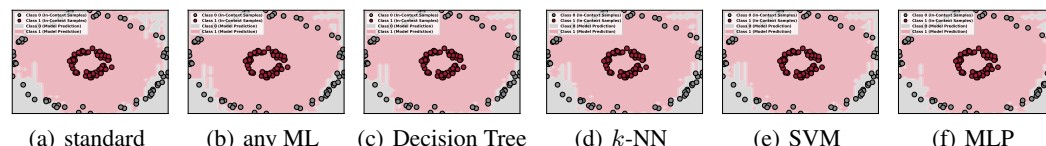

(a) standard    (b) any ML    (c) Decision Tree    (d) $k$-NN    (e) SVM    (f) MLP

Figure 21: **Decision boundaries of Llama-3-8B on the circle classification with implicit reasoning.** The prompts include both the standard prompt and the prompts where learning strategies are specified. We can observe that Llama-3-8B fails to correctly predict the labels of query data.

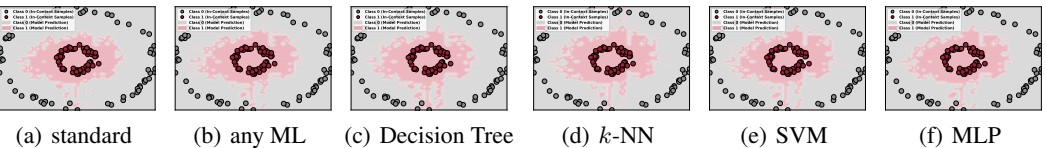

(a) standard    (b) any ML    (c) Decision Tree    (d) $k$-NN    (e) SVM    (f) MLP

Figure 22: **Decision boundaries of Mistral-8B on the circle classification task with implicit reasoning.** The prompts include both the standard prompt and prompts where learning strategies are specified. According to figures, we can observe that Mistral-8B can generate decision boundaries in a similar way to conventional ML, although the decision boundaries remain fragmented.

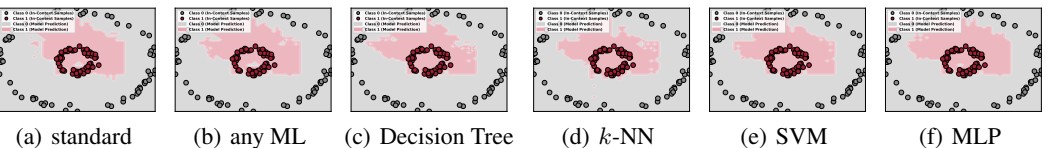

(a) standard    (b) any ML    (c) Decision Tree    (d) $k$-NN    (e) SVM    (f) MLP

Figure 23: **Decision boundaries of Qwen-2.5-7B on the circle classification task with implicit reasoning.** The prompts include both the standard prompt and the prompts where learning strategies are specified. According to these figures, we can observe that Qwen-2.5-7B can generate decision boundaries in a similar way to conventional ML, although the boundaries remain fragmented.

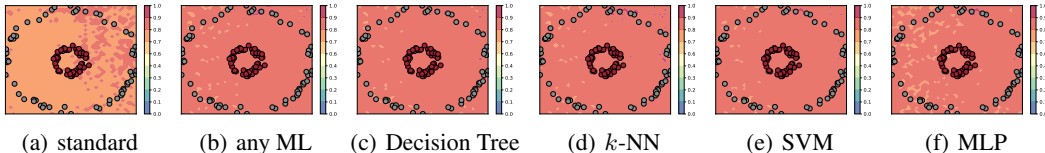

(a) standard    (b) any ML    (c) Decision Tree    (d) $k$-NN    (e) SVM    (f) MLP

Figure 24: **Confidence of predictions of Llama-3-8B on the circle classification task with implicit reasoning.** The prompts include both the standard prompt and prompts where learning strategies are specified.

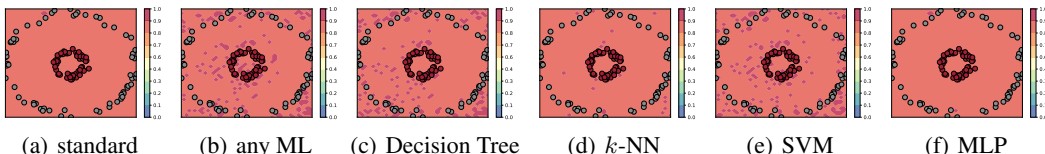

(a) standard    (b) any ML    (c) Decision Tree    (d) $k$-NN    (e) SVM    (f) MLP

Figure 25: **Confidence of predictions of Mistral-8B on the circle classification task with implicit reasoning.** The prompts include both the standard prompt and the prompts where learning strategies are specified.

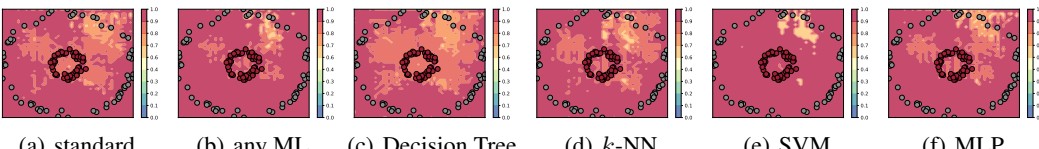

(a) standard    (b) any ML    (c) Decision Tree    (d) $k$-NN    (e) SVM    (f) MLP

Figure 26: **Confidence of predictions of Qwen-2.5-7B on the circle classification task with implicit reasoning.** The prompts include both the standard prompt and the prompts where learning strategies are specified.

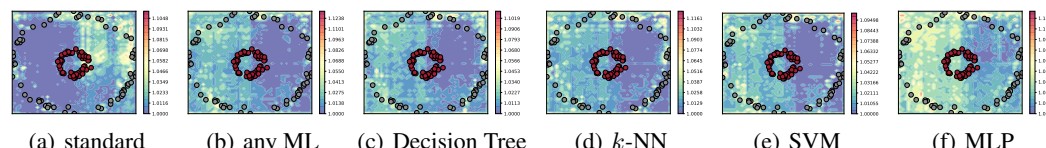

(a) standard      (b) any ML      (c) Decision Tree      (d) $k$-NN      (e) SVM      (f) MLP

Figure 27: **Perplexities of predictions of Llama-3-8B on the circle classification task with implicit reasoning.** The prompts include both the standard prompt and the prompts where learning strategies are specified.

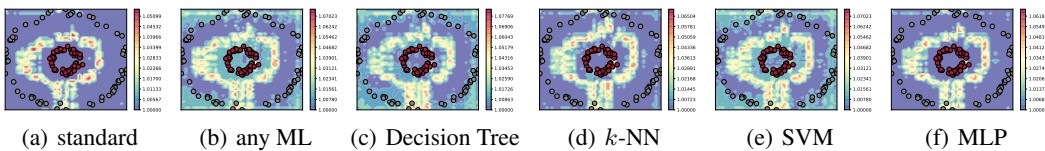

(a) standard      (b) any ML      (c) Decision Tree      (d) $k$-NN      (e) SVM      (f) MLP

Figure 28: **Perplexities in predictions of Mistral-8B on the circle classification task with implicit reasoning.** The prompts include both the standard prompt and the prompts where learning strategies are specified.

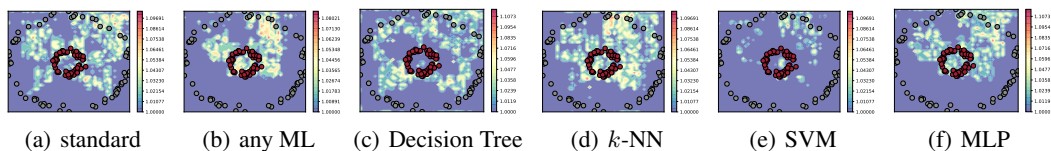

(a) standard      (b) any ML      (c) Decision Tree      (d) $k$-NN      (e) SVM      (f) MLP

Figure 29: **Perplexities in predictions of Qwen-2.5-7B on the circle classification task with implicit reasoning.** The prompts include both the standard prompt and the prompts where learning strategies are specified.

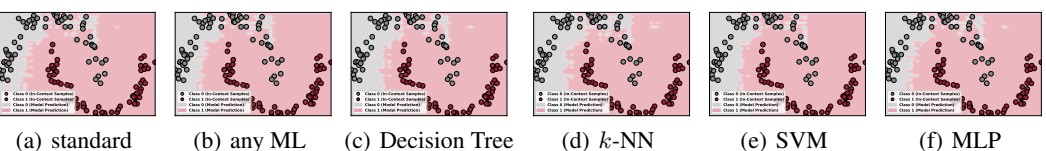

(a) standard      (b) any ML      (c) Decision Tree      (d) $k$-NN      (e) SVM      (f) MLP

Figure 30: **Decision boundaries of Llama-3-8B on the moon classification with implicit reasoning. The prompts include both the standard prompt and the prompts where learning strategies are specified.** We can observe that Llama-3-8B fails to correctly predict the labels of query data.

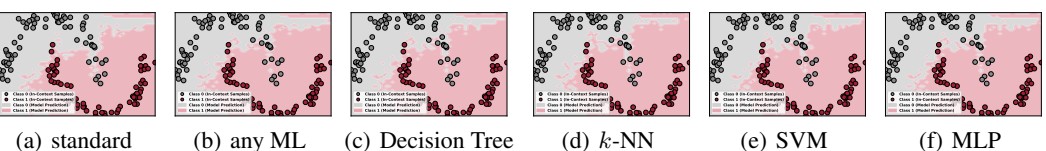

(a) standard      (b) any ML      (c) Decision Tree      (d) $k$-NN      (e) SVM      (f) MLP

Figure 31: **Decision boundaries of Mistral-8B on the moon classification task with implicit reasoning. The prompts include both the standard prompt and prompts where learning strategies are specified.** According to figures, we can observe that Mistral-8B can generate decision boundaries in a similar way to conventional ML, although the decision boundaries remain fragmented.

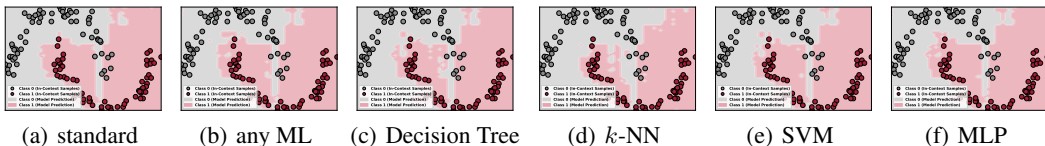

(a) standard      (b) any ML      (c) Decision Tree      (d) $k$-NN      (e) SVM      (f) MLP

Figure 32: **Decision boundaries of Qwen-2.5-7B on the moon classification task with implicit reasoning. The prompts include both the standard prompt and the prompts where learning strategies are specified.** According to these figures, we can observe that Qwen-2.5-7B can generate decision boundaries in a similar way to conventional ML, although the boundaries remain fragmented.

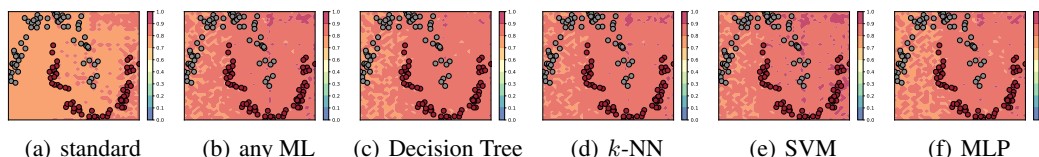

(a) standard     (b) any ML     (c) Decision Tree     (d) $k$-NN     (e) SVM     (f) MLP

Figure 33: **Confidence of predictions of Llama-3-8B on the moon classification task with implicit reasoning.** The prompts include both the standard prompt and prompts where learning strategies are specified.

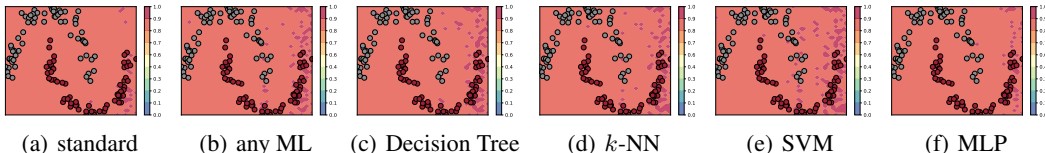

(a) standard     (b) any ML     (c) Decision Tree     (d) $k$-NN     (e) SVM     (f) MLP

Figure 34: **Confidence of predictions of Mistral-8B on the moon classification task with implicit reasoning.** The prompts include both the standard prompt and the prompts where learning strategies are specified.

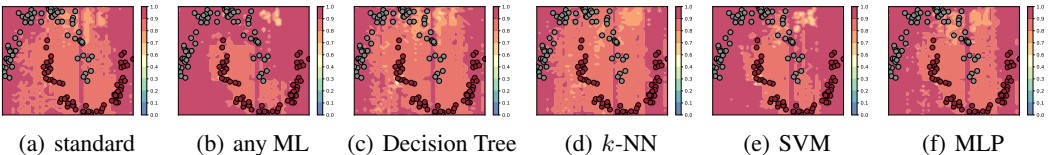

(a) standard     (b) any ML     (c) Decision Tree     (d) $k$-NN     (e) SVM     (f) MLP

Figure 35: **Confidence of predictions of Qwen-2.5-7B on the moon classification task with implicit reasoning.** The prompts include both the standard prompt and the prompts where learning strategies are specified.

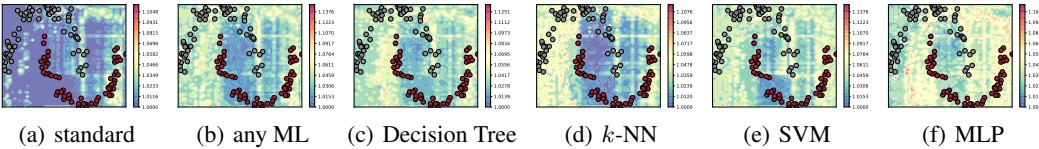

(a) standard     (b) any ML     (c) Decision Tree     (d) $k$-NN     (e) SVM     (f) MLP

Figure 36: **Perplexities of predictions of Llama-3-8B on the moon classification task with implicit reasoning.** The prompts include both the standard prompt and the prompts where learning strategies are specified.

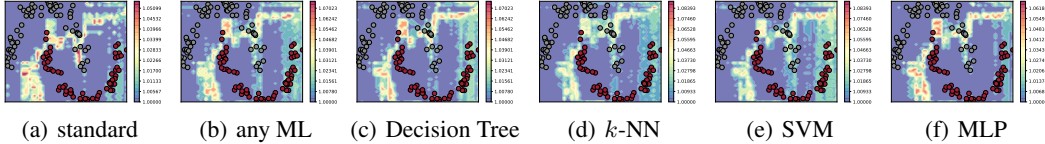

(a) standard     (b) any ML     (c) Decision Tree     (d) $k$-NN     (e) SVM     (f) MLP

Figure 37: **Perplexities in predictions of Mistral-8B on the moon classification task with implicit reasoning.** The prompts include both the standard prompt and the prompts where learning strategies are specified.

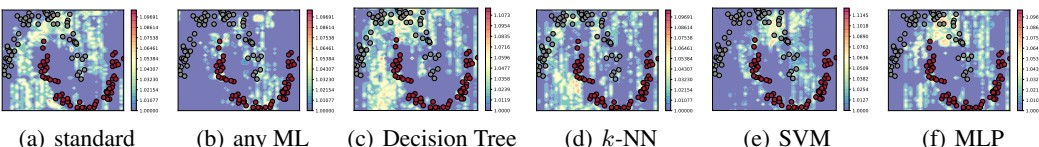

(a) standard     (b) any ML     (c) Decision Tree     (d) $k$-NN     (e) SVM     (f) MLP

Figure 38: **Perplexities in predictions of Qwen-2.5-7B on the moon classification task with implicit reasoning.** The prompts include both the standard prompt and the prompts where learning strategies are specified.

### D.3 DECISION BOUNDARIES OF EXPLICIT REASONING ON MORE LLMS AND CLASSIFICATION TASKS

In Section 4.3, we have explored that explicit reasoning, in both cases where the standard prompt and the prompts with specified learning strategies are applied, the LLMs (i.e., Llama-3-8B and Qwen-2.5-7B) fail to achieve smooth decision boundaries. Compared to implicit reasoning, explicit reasoning obtains even worse decision boundaries. To further determine whether this phenomenon widely exists among all LLMs, we propose to validate this phenomenon with more LLMs and classification tasks.

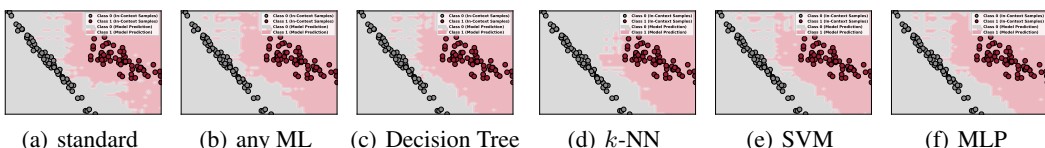

(a) standard      (b) any ML      (c) Decision Tree      (d) $k$-NN      (e) SVM      (f) MLP

Figure 39: **Decision boundaries of Qwen-2.5-7B in the linear classification task with explicit reasoning.** The figures show the decision boundaries of Qwen-2.5-7B with both the standard prompt and the prompts where machine learning strategies are specified. According to these figures, we can observe that the decision boundaries are not changed significantly compared to the standard case.

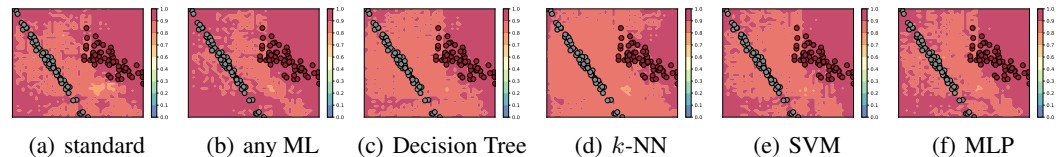

(a) standard      (b) any ML      (c) Decision Tree      (d) $k$-NN      (e) SVM      (f) MLP

Figure 40: **Confidence of predictions of Qwen-2.5-7B in the linear classification task with explicit reasoning.** The figures show confidence of Qwen-2.5-78B in predictions of query data in the way of implicit reasoning with both the standard prompt and the prompts where machine learning strategies are specified. The results reveal that confidence increases when learning strategies are specified.

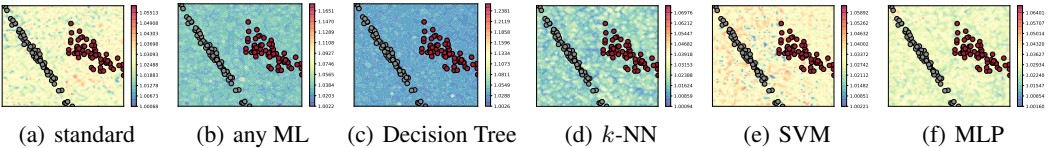

(a) standard      (b) any ML      (c) Decision Tree      (d) $k$-NN      (e) SVM      (f) MLP

Figure 41: **Perplexities in predictions of Qwen-2.5-7B in the linear classification task with explicit reasoning.** The figures show the perplexities of Qwen-2.5-7B in its predictions in the linear classification task in the way of implicit reasoning with both the standard prompt and prompts where machine learning strategies are specified. The low perplexity indicates high confidence of LLMs.

#### D.3.1 QWEN-2.5-7B ON LINEAR CLASSIFICATION TASK WITH ER

The visualization results of decision boundaries, confidence, and perplexity of Qwen-2.5-7B on the linear classification task with implicit reasoning are reported in Figs. 39, 40, and 41, respectively. According to the visualization results, we can observe the following phenomena. (1) Consistent with Llama-3-8B, the decision boundaries derived from Qwen-2.5-7B are also fragmented in all cases. Consistent with the implicit reasoning case, when learning strategies are specified, the decision boundaries are improved slightly (e.g., $k$-NN). (2) For the confidence, we can observe that Qwen-2.5-7B is quite confident in its predictions, although such high confidence does not help improve the decision boundaries. Moreover, consistent with implicit reasoning, we also notice that the confidence changes in a random way when learning strategies are specified in prompts. Specifically, when learning strategies, such as Decision Tree (Fig. 40(c)) and $k$-NN (Fig. 40(d)), are specified in prompts, the confidence decreases significantly. However, when Qwen-2.5-7B is allowed to leverage any machine learning methods (Fig. 40(b)), the confidence increases slightly. (3) For the perplexity, Qwen-2.5-7B reacts differently towards different learning strategies. Specifically, we can observe that the perplexity increases when SVM and MLP are specified in prompts. However, when arbitrary machine learning strategies, Decision Tree, and $k$-NN are allowed, the perplexity obviously decreases.

### D.3.2 DECISION BOUNDARIES ON NON-LINEAR CLASSIFICATION TASKS WITH ER

In this section, we further explore the in-context learning capability of LLMs (Llama-3-8B and Qwen-2.5-7B) on the non-linear classification tasks (i.e., circle classification task and moon classification task). The visualization results of decision boundaries, confidence, and perplexity of all LLMs with implicit reasoning prompted by different types of prompts are respectively reported in Figs. 42-53.

From the perspective of decision boundaries, in both circle and moon classification tasks, the decision boundaries derived from Llama-3-8B and Qwen-2.5-7B are fragmented (see Figs. 42-43 and Figs. 48-49). However, compared to Llama-3-8B, the decision boundaries derived from Qwen-2.5-7B are clearer. Such a phenomenon is consistent with that observed in implicit reasoning. Meanwhile, we can also observe that the decision boundaries are not improved when learning strategies are specified.

For confidence, the phenomena are consistent with those in the linear classification task. Specifically, for Llama-3-8B, specifying learning strategies increases the confidence of predictions in all cases. However, for Qwen-2.5-7B, the LLM reacts differently to different learning strategies. The phenomenon exists in both circle (see Figs. 44-45) and moon classification tasks (see Figs. 50-51).

For the perplexity, in both circle and moon classification tasks, we can observe that both LLMs react differently to different learning strategies. However, different from the cases of implicit reasoning, specifying learning strategies can holistically lower the perplexity of Qwen-2.5-7B in some cases. For example, in the moon classification task, when strategies, such as any ML (Fig. 53(b)), Decision Tree (Fig. 53(c)), and $k$-NN (Fig. 53(d)), are applied, the perplexity of Qwen-2.5-7B decreases holistically.

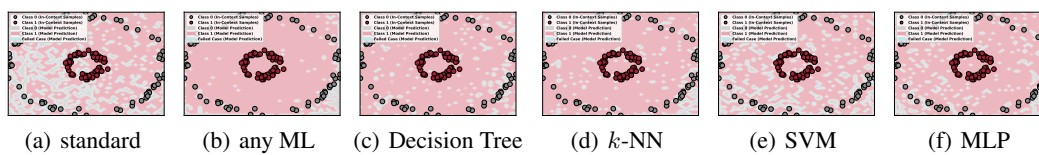

(a) standard     (b) any ML     (c) Decision Tree     (d) $k$-NN     (e) SVM     (f) MLP

Figure 42: **Decision boundaries of Llama-3-8B on the circle classification with explicit reasoning.** The prompts include both the standard prompt and the prompts where learning strategies are specified.

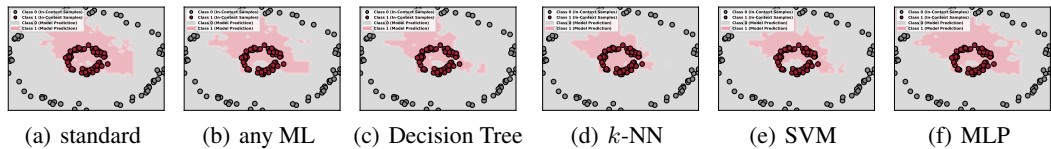

(a) standard     (b) any ML     (c) Decision Tree     (d) $k$-NN     (e) SVM     (f) MLP

Figure 43: **Decision boundaries of Qwen-2.5-7B on the circle classification task with explicit reasoning.** The prompts include both the standard prompt and the prompts where learning strategies are specified.

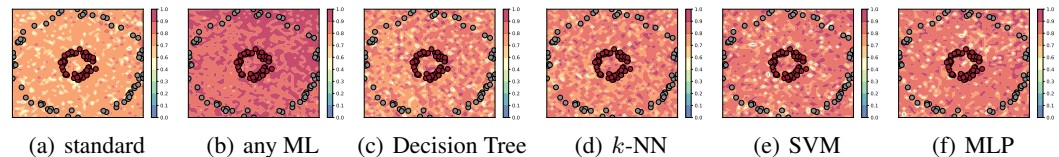

(a) standard     (b) any ML     (c) Decision Tree     (d) $k$-NN     (e) SVM     (f) MLP

Figure 44: **Confidence of predictions of Llama-3-8B on the circle classification task with explicit reasoning.** The prompts include both the standard prompt and prompts where learning strategies are specified.

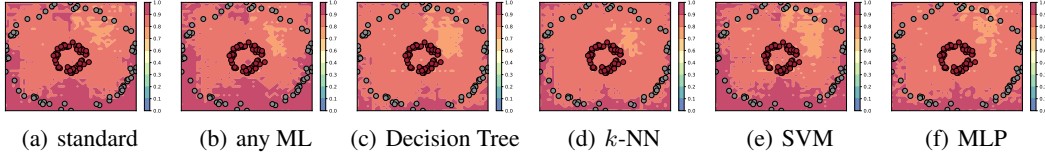

(a) standard     (b) any ML     (c) Decision Tree     (d) $k$-NN     (e) SVM     (f) MLP

Figure 45: **Confidence of predictions of Qwen-2.5-7B on the circle classification task with explicit reasoning.** The prompts include both the standard prompt and the prompts where learning strategies are specified.

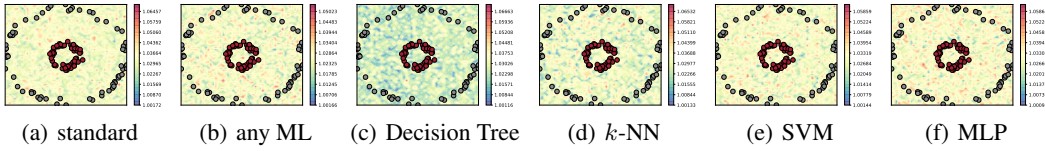

(a) standard     (b) any ML     (c) Decision Tree     (d) $k$-NN     (e) SVM     (f) MLP

Figure 46: **Perplexities of predictions of Llama-3-8B on the circle classification task with explicit reasoning.** The prompts include both the standard prompt and the prompts where learning strategies are specified.

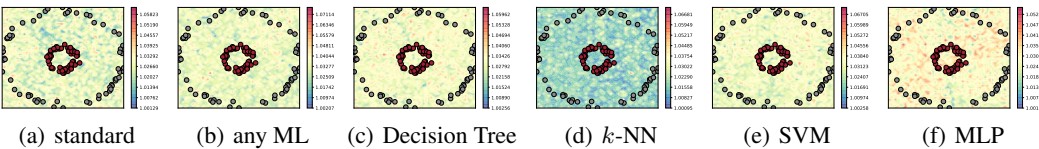

(a) standard     (b) any ML     (c) Decision Tree     (d) $k$-NN     (e) SVM     (f) MLP

Figure 47: **Perplexities in predictions of Qwen-2.5-7B on the circle classification task with explicit reasoning.** The prompts include both the standard prompt and the prompts where learning strategies are specified.

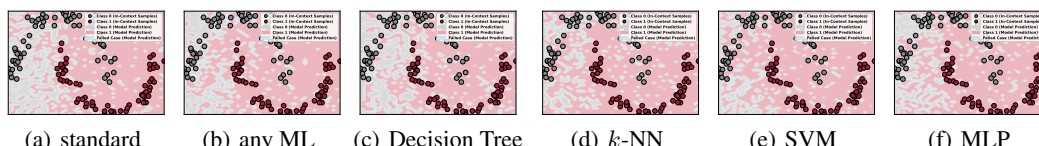

(a) standard (b) any ML (c) Decision Tree (d) $k$-NN (e) SVM (f) MLP

Figure 48: **Decision boundaries of Llama-3-8B on the moon classification with explicit reasoning. The prompts include both the standard prompt and the prompts where learning strategies are specified.** We can observe that Llama-3-8B fails to correctly predict the labels of query data.

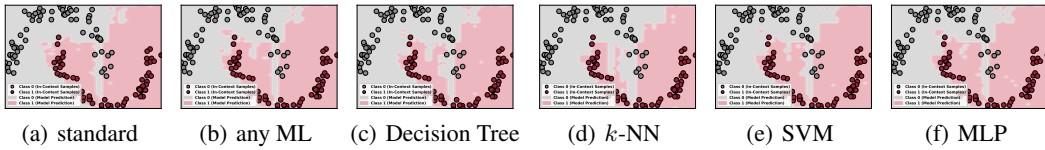

(a) standard (b) any ML (c) Decision Tree (d) $k$-NN (e) SVM (f) MLP

Figure 49: **Decision boundaries of Qwen-2.5-7B on the moon classification task with explicit reasoning. The prompts include both the standard prompt and the prompts where learning strategies are specified.** According to these figures, we can observe that Qwen-2.5-7B can generate decision boundaries in a similar way to conventional ML, although the boundaries remain fragmented.

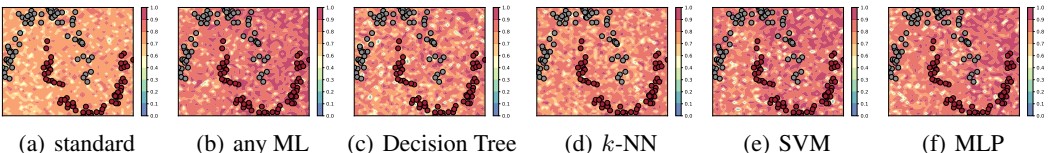

(a) standard (b) any ML (c) Decision Tree (d) $k$-NN (e) SVM (f) MLP

Figure 50: **Confidence of predictions of Llama-3-8B on the moon classification task with explicit reasoning.** The prompts include both the standard prompt and prompts where learning strategies are specified.

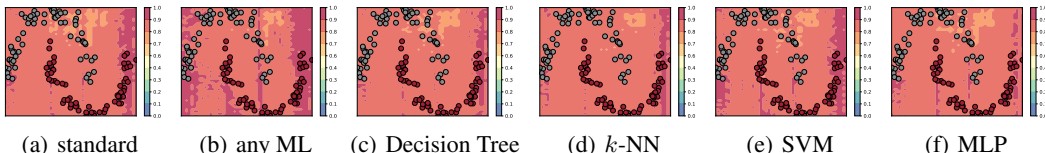

(a) standard (b) any ML (c) Decision Tree (d) $k$-NN (e) SVM (f) MLP

Figure 51: **Confidence of predictions of Qwen-2.5-7B on the moon classification task with explicit reasoning.** The prompts include both the standard prompt and the prompts where learning strategies are specified.

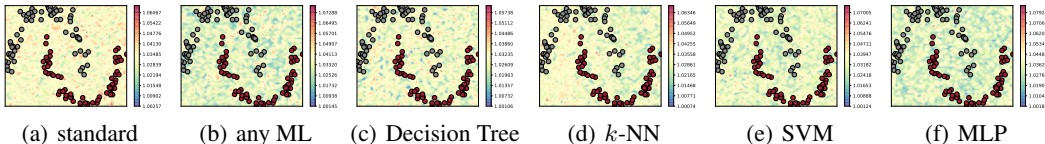

(a) standard (b) any ML (c) Decision Tree (d) $k$-NN (e) SVM (f) MLP

Figure 52: **Perplexities of predictions of Llama-3-8B on the moon classification task with explicit reasoning.** The prompts include both the standard prompt and the prompts where learning strategies are specified.

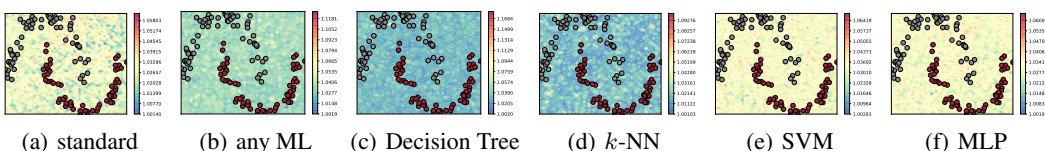

(a) standard (b) any ML (c) Decision Tree (d) $k$-NN (e) SVM (f) MLP

Figure 53: **Perplexities in predictions of Qwen-2.5-7B on the moon classification task with explicit reasoning.** The prompts include both the standard prompt and the prompts where learning strategies are specified.

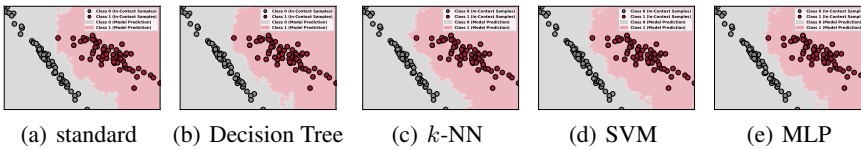

(a) standard     (b) Decision Tree     (c) $k$-NN     (d) SVM     (e) MLP

Figure 54: **Decision boundaries of DeepSeek-V3 with implicit reasoning where learning strategies are specified.** The figures show the decision boundaries of DeepSeek-V3 with prompts where machine learning strategies are specified. According to these figures, we can observe that the positions of decision boundaries change slightly and the boundaries are improved slightly in some cases.

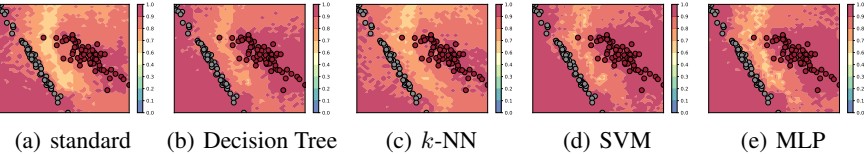

(a) standard     (b) Decision Tree     (c) $k$-NN     (d) SVM     (e) MLP

Figure 55: **Confidence of predictions of DeepSeek-V3 with implicit reasoning where learning strategies are specified.** The figures show the confidence of DeepSeek-V3 in its predictions of query data with implicit reasoning with the prompts where machine learning strategies are specified.

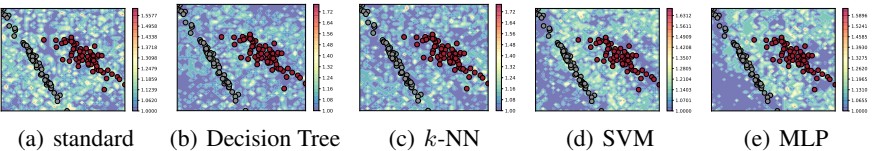

(a) standard     (b) Decision Tree     (c) $k$-NN     (d) SVM     (e) MLP

Figure 56: **Perplexities in predictions of DeepSeek-V3 in the linear classification task with implicit reasoning.** The figures show the perplexities of DeepSeek-V3 in its predictions in the linear classification task in the way of implicit reasoning with both the standard prompt and prompts where machine learning strategies are specified.

### D.4 DEEPSEEK-V3 ON LINEAR CLASSIFICATION TASK

In previous sections, we have evaluated both linear and non-linear classification tasks on small open-source LLMs, such as Llama, Qwen, and Mistral. However, whether these phenomena are consistent on more powerful LLMs remains unclear. To further figure this out, we evaluate DeepSeek-V3 (Liu et al., 2024) on the linear classification task with both implicit and explicit reasoning methods.

#### D.4.1 DEEPSEEK-V3 ON LINEAR CLASSIFICATION TASK WITH IR

Following the previous cases, we investigate the decision boundaries, confidence, and perplexity of DeepSeek-V3 respectively with the standard prompt and prompts where machine learning methods are specified. The results are visualized in Figs. 54, 55, and 56. According to Fig. 54, we can observe that the decision boundaries derived from DeepSeek-V3 are consistently fragmented with the previous open-source LLMs and specifying machine learning methods does not improve the decision boundaries. Moreover, according to Fig. 55, we can also observe that specifying machine learning methods in prompts significantly increases the confidence of DeepSeek-V3 in its predictions. Different from Llama and Mistral, specifying machine learning methods in prompts also lower down the perplexity of DeepSeek-V3 in its predictions (Fig. 56). Specifically, with machine learning methods specified in prompts, the low perplexity areas converge to the regions of the two data clusters.

#### D.4.2 DEEPSEEK-V3 ON LINEAR CLASSIFICATION TASK WITH ER

In this section, we further investigate the decision boundaries, confidence, and perplexity of DeepSeek-V3 respectively with the standard prompt and prompts where machine learning methods are specified. The results are visualized in Figs. 57, 58, and 59. According to Fig. 57, the decision boundaries are more fragmented than those derived with implicit reasoning. Moreover, specifying machine learning methods in prompts even worsens the situation. These phenomena are consistent with those found in Qwen, Mistral, and Llama. Moreover, we can also observe that, in most cases, specifying machine learning methods in prompts enhances the confidence and lowers the perplexity of DeepSeek-V3.

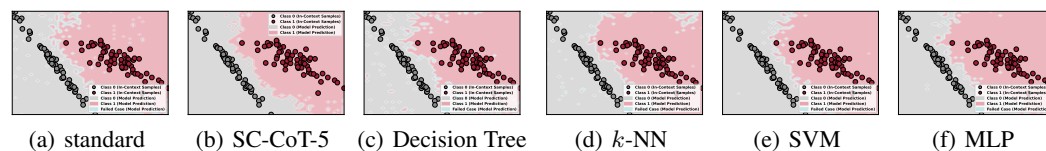

(a) standard  (b) SC-CoT-5  (c) Decision Tree  (d) $k$-NN  (e) SVM  (f) MLP

Figure 57: **Decision boundaries of DeepSeek-V3 with explicit reasoning where learning strategies are specified.** The figures show the decision boundaries of DeepSeek-V3 with prompts where machine learning strategies are specified.

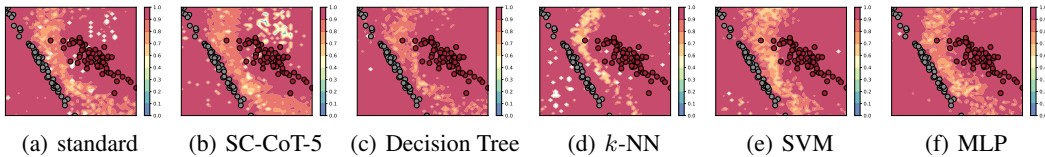

(a) standard  (b) SC-CoT-5  (c) Decision Tree  (d) $k$-NN  (e) SVM  (f) MLP

Figure 58: **Confidence of predictions of DeepSeek-V3 with explicit reasoning where learning strategies are specified.** The figures show the confidence of DeepSeek-V3 in its predictions of query data with explicit reasoning with the prompts where machine learning strategies are specified.

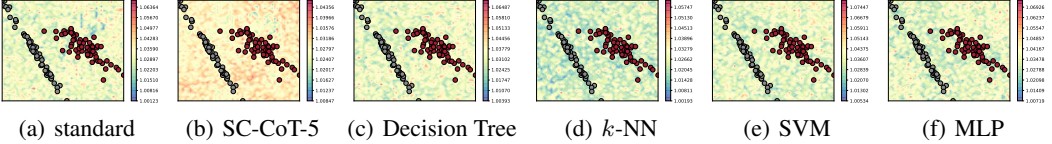

(a) standard  (b) SC-CoT-5  (c) Decision Tree  (d) $k$-NN  (e) SVM  (f) MLP

Figure 59: **Perplexities in predictions of DeepSeek-V3 in the linear classification task with explicit reasoning.** The figures show the perplexities of DeepSeek-V3 in its predictions in the linear classification task with explicit reasoning with both the standard prompt and prompts where machine learning strategies are specified.

However, unlike small open-source LLMs, performing reasoning in the way of SC-CoT (e.g., SC-CoT-5) does not help improve decision boundaries. Meanwhile, the application of CoT further increases the perplexity of DeepSeek-V3. Such a phenomenon, to some extent, indicates that the larger LLMs may already be good enough in performing reasoning on the classification task, and strategies for enhancing the reasoning cannot provide any further improvements.

### D.5 INVESTIGATION OF HIGH-DIMENSIONAL CLASSIFICATION TASK

In the main paper, we have discussed the decision boundaries derived from Llama-3-8B with both implicit and explicit reasoning. Generally, there are three main points worth noticing. Firstly, the decision boundaries derived from LLMs are all fragmented. Secondly, the decision boundaries derived with implicit reasoning are better than those derived from explicit reasoning. Thirdly, the decision boundaries derived from reasoning with the prompts where learning strategies are specified are worse than those derived from reasoning with the standard prompt. All these aspects are summarized from the experiments in the 2D linear classification task. In order to validate the generality of these findings, in this section, we propose to evaluate LLMs on high-dimensional linear classification tasks.

For simplicity, in this section, we evaluate Llama-3-8B in the 3D linear classification task. The decision boundaries are measured with our proposed smooth score. The results of the smooth score are reported in Table 5. According to the results in the table, we can observe that (1) the decision boundaries are fragmented since the smooth scores derived from Llama-3-8B with both implicit and explicit reasoning are smaller than those obtained from conventional machine learning methods; (2) compared to explicit reasoning, the decision boundaries obtained with implicit reasoning are much better since the smooth scores are relatively high; and (3) the decision boundary derived with prompts, where learning strategies are specified, is worse than that derived with the standard prompt in implicit reasoning. However, in the explicit reasoning, the decision boundary derived from LLM prompted with the standard prompt is worse than that derived from Llama-3-8B prompted with the prompts where learning strategies are specified. Overall, we can observe that the phenomena found in the high-dimensional classification task are consistent with those in the 2D linear classification task.

Table 5: Smooth scores of decision boundaries of both conventional ML methods (scikit-learn) and (implicit/explicit) reasoning of LLMs in the 3D linear classification task (top-10 nearest neighbours).

| Case | Standard | Any ML | Decision Tree | $k$-NN | MLP | SVM |
|------|----------|--------|---------------|--------|-----|-----|
| Baseline (sklearn) | - | - | 0.9685 | 0.9570 | 0.9671 | 0.9370 |
| Llama-3-8B (IR) | 0.8750 | 0.8491 | 0.8661 | 0.8765 | 0.8715 | 0.8546 |
| Llama-3-8B (ER) | 0.6886 | 0.8280 | 0.6153 | 0.7494 | 0.7826 | 0.7179 |

## E  ROBUSTNESS OF SMOOTH SCORE

In Section 4.1, we propose a metric, smooth score (cf. Eq. (2)), to measure the smoothness of decision boundaries derived from LLMs. The insight of the proposed smooth score is that each query sample (except for the query samples at the decision boundary) shares the same prediction result as other samples in a small neighborhood. Note that the smooth score only cares about the smoothness of decision boundaries, while the correctness of the predictions is not considered in our case. Since the query data are derived from the plane where the in-context data lie, we do not have the well-calibrated labels for measuring the accuracy of the predictions. Thus, even in the case where decision boundaries obtained from LLMs are badly placed, the smooth decision boundaries can still achieve high scores.

**Sensitiveness of the choice of $k$.** Since calculating the smooth score is closely related to the $k$ predictions of query data in the neighborhood, the smooth score can be treated as a $k$-NN agreement rate, which may be sensitive to the choice of $k$. In order to determine the robustness of the smooth score to the hyperparameter $k$, we propose to calculate the smooth score with different $k$ values. Specifically, we calculate the smooth scores of the prediction results of Llama-3-8B on the 2D linear classification task under the settings of implicit reasoning, respectively, with $k = \{3, 4, 10, 100\}$.

The results are reported in Table 6. According to the table, we can observe that the smooth score consistently reflects the smoothness of decision boundaries in Figs. 1 and 3. Specifically, the smooth scores of decision boundaries obtained from the sklearn tools are higher than those derived from LLMs, which is consistent with the visualization results that sklearn achieves better decision boundaries than LLMs. Moreover, we can also observe that the decision boundary generated from the standard settings achieves a better smooth score than the cases where machine learning strategies are specified when $k$ is not so large. However, when $k$ is quite large (e.g., 100), the case changes. Thus, we can summarize that the smooth score is robust to $k$ when $k$ varies in a reasonable range.

Table 6: Results of smooth score with different $k$ values on the 2D linear classification task with Llam-3-8B under the settings of the implicit reasoning paradigm. In this case, we mainly examine the smooth score with $k = \{3, 4, 10, 100\}$.

| Case | Standard | Any ML | Decision Tree | $k$-NN | MLP | SVM |
|---|---|---|---|---|---|---|
| Baseline (sklearn) | - | - | 0.9868 | 0.9854 | 0.9854 | 0.9830 |
| Llama-3-8B (IR, $k = 3$) | 0.9826 | 0.9802 | 0.9815 | 0.9799 | 0.9794 | 0.9810 |
| Llama-3-8B (IR, $k = 4$) | 0.9772 | 0.9764 | 0.9768 | 0.9758 | 0.9758 | 0.9750 |
| Llama-3-8B (IR, $k = 10$) | 0.9735 | 0.9730 | 0.9733 | 0.9734 | 0.9722 | 0.9723 |
| Llama-3-8B (IR, $k = 100$) | 0.9433 | 0.9436 | 0.9440 | 0.9442 | 0.9447 | 0.9435 |

**Limitation of smooth score.** Another concern regarding our proposed smooth score is that the nearest-neighbor signals fade quickly when the dimension of the space increases. Moreover, we also notice that increasing the space dimension will result in the exponential increase of the amount of query data, which, in turn, increases the cost of reasoning. Thus, as a trade-off, we conduct our high-dimensional experiment in 3D space. Thus, a potential limitation of the smooth score is the space dimension. Specifically, the proposed smooth score cannot be applied to very high dimensions.

## F  DETAILED DISCUSSION OF BEHAVIORS OF IMPLICIT REASONING OF LLMS

### F.1  EXPLORATION OF UNDERLYING BEHAVIOR OF IMPLICIT REASONING OF LLMS

In Section 5, we propose to explore the underlying behavior of the implicit reasoning of Llama-3-8B in the linear classification task. To this end, we propose a method, which measures the difference between the KL-divergence score of a set of LLM predictions and a set of predictions of conventional machine learning methods and the KL-divergence of a set of predictions of conventional machine learning methods themselves. Briefly, if the difference between the two KL-divergence scores is small, we can then believe that the behaviors of LLMs and conventional ML methods are the same.

In the main paper, we mainly consider the linear classification task on Llama-3-8B. According to the quantitative results in Table 4, we find that the difference between the two KL-divergence ($\bar{d}_{\text{LLM}\rightarrow\text{method}}$ & $\bar{d}_{\text{Conv}}$ in "Standard" case) is small, which demonstrates that Llama-3-8B behaves similarly to conventional machine learning methods when performing implicit reasoning. Meanwhile, when learning strategies are specified (i.e., Decision Tree (LLM), $k$-NN (LLM), MLP (LLM), and SVM (LLM)), the difference between the two KL-divergence scores becomes larger, which indicates that Llama-3-8B fails to behave in the same way as conventional machine learning methods. These results, to some extent, can be used to explain why specifying learning strategies does not help improve decision boundaries. The observations can be supported by the visualizations in Figs. 8 (Standard), 63 (Decision Tree (LLM)), 66 ($k$-NN (LLM)), 69 (MLP (LLM)), and 72 (SVM (LLM)).

For other tasks, such as the circle and the moon classification task, the gaps between $\bar{d}_{\text{LLM}\rightarrow\text{method}}$ and $\bar{d}_{\text{Conv}}$ of Llama-3-8B are quite large (see Tables 7 & 8), which indicates that Llama-3-8B fails to perform prediction in the same way as conventional machine learning methods. The findings are also supported by the visualization results in Figs. 9 & 62. However, for Mistral-8B (Tables 10-11) and Qwen-2.5-7B (Tables 13-14), although the obtained decision boundaries are not so good as conventional machine learning methods, better decision boundaries can be obtained than Llama-3-8B.

With all these aspects taken into consideration, we can summarize that the intrinsic capability of LLMs performing in-context learning in the way of implicit reasoning resembles the behavior of conventional machine learning methods. However, such a capability is limited by two aspects. On the one hand, the capability is fragile, it may be damaged by specific instructions in prompts, such as the specified learning strategies. On the other hand, the capability is constrained by the pretraining. Specifically, for LLMs pretrained later, such as Mistral-8B and Qwen-2.5-7B, they can perform both simple and complex classification tasks, ranging from the linear classification task to the moon classification task. However, for LLMs (e.g., Llama-3-8B), which are pretrained earlier, the capability can merely be used to solve simple classification tasks. For a comprehensive exploration of this phenomenon, we provide more quantitative and visualization results in following tables and figures.

### F.2  DISTRIBUTION DISCREPANCIES MATTERS IN DECISION BOUNDARIES

Another aspect that is worth noticing is the distribution discrepancies between the in-context data and the query data. Specifically, in the previous work (Zhao et al., 2024), the query data are generated by uniformly splitting the plane, where the in-context data lie, into several grids. In this case, both in-distribution and out-of-distribution data are included. However, in the experimental setting of this section, the in-context data and the query data are generated from the same distribution in a space. Such a difference provides us an opportunity to explore whether the decision boundaries derived from LLMs are influenced by the distribution discrepancies between the in-context data and the query data.

Specifically, according to the visualized prediction results of classification tasks, where in-context and query data are sampled from the same distribution, we can observe that LLMs behave similarly to conventional machine learning methods in tasks, although their performance is predefined during the pre-training phase. For example, the quality of the data used in pretraining will affect the performance of LLMs. Compared to the fragmented decision boundaries and the dissimilar behaviors of LLMs in previous sections, such a phenomenon indicates that LLMs can perform better on in-distribution data and tend to perform classification in a similar way to conventional machine learning methods. This further implies that the distribution discrepancy between the in-context data and the query data plays an important role in performing predictions and will significantly influence the decision boundaries. Such a phenomenon is consistent with that observed in the generalization problem of conventional machine learning. Thus, we can achieve a sense that data distribution matters in decision boundaries.

Table 7: **The KL-divergence results of predictions on the circle classification task, respectively, between Llama-3-8B and itself, conventional machine learning algorithms and themselves, and Llama-3-8B and conventional machine learning algorithms.** $d_{\text{LLM}}$ denotes the KL-divergence of predictions between Llama-3-8B and itself, $d_{\text{Conv}}$ denotes the KL-divergence of predictions of conventional machine learning and themselves, and $d_{\text{LLM}\to\text{method}}$ denotes the KL-divergence of predictions between Llama-3-8B and conventional ML methods, where $\text{method} \in \{\text{DT}, \text{kNN}, \text{MLP}, \text{SVM}\}$.

| Case | $\bar{d}_{\text{LLM}}$ | $\bar{d}_{\text{Conv}}$ | $\bar{d}_{\text{LLM}\to\text{DT}}$ | $\bar{d}_{\text{LLM}\to\text{kNN}}$ | $\bar{d}_{\text{LLM}\to\text{MLP}}$ | $\bar{d}_{\text{LLM}\to\text{SVM}}$ |
|---|---|---|---|---|---|---|
| Standard (LLM) | 1.77 | - | 2.33 | 2.31 | 2.32 | 2.32 |
| Decision Tree (LLM) | 1.54 | 7.64 | 2.14 | 2.11 | 2.12 | 2.12 |
| $k$-NN (LLM) | 1.91 | 7.34 | 2.47 | 2.44 | 2.45 | 2.45 |
| MLP (LLM) | 1.72 | 7.56 | 2.30 | 2.27 | 2.28 | 2.28 |
| SVM (LLM) | 1.60 | 7.54 | 2.19 | 2.15 | 2.17 | 2.16 |

Table 8: **The KL-divergence results of predictions on the moon classification task, respectively, between Llama-3-8B and itself, conventional machine learning algorithms and themselves, and Llama-3-8B and conventional machine learning algorithms.** $d_{\text{LLM}}$ denotes the KL-divergence of predictions between Llama-3-8B and itself, $d_{\text{Conv}}$ denotes the KL-divergence of predictions of conventional machine learning and themselves, and $d_{\text{LLM}\to\text{method}}$ denotes the KL-divergence of predictions between Llama-3-8B and conventional ML methods, where $\text{method} \in \{\text{DT}, \text{kNN}, \text{MLP}, \text{SVM}\}$.

| Case | $\bar{d}_{\text{LLM}}$ | $\bar{d}_{\text{Conv}}$ | $\bar{d}_{\text{LLM}\to\text{DT}}$ | $\bar{d}_{\text{LLM}\to\text{kNN}}$ | $\bar{d}_{\text{LLM}\to\text{MLP}}$ | $\bar{d}_{\text{LLM}\to\text{SVM}}$ |
|---|---|---|---|---|---|---|
| Standard (LLM) | 3.54 | - | 3.99 | 3.95 | 3.96 | 3.95 |
| Decision Tree (LLM) | 3.80 | 7.80 | 4.23 | 4.20 | 4.20 | 4.20 |
| $k$-NN (LLM) | 4.20 | 7.61 | 4.62 | 4.58 | 4.58 | 4.58 |
| MLP (LLM) | 3.84 | 7.59 | 4.28 | 4.24 | 4.24 | 4.24 |
| SVM (LLM) | 3.50 | 7.61 | 3.95 | 3.92 | 3.92 | 3.92 |

Table 9: **The KL-divergence results of predictions on the linear classification task, respectively, between Mistral-8B and itself, conventional machine learning algorithms and themselves, and Mistral-8B and conventional machine learning algorithms.** $d_{\text{LLM}}$ denotes the KL-divergence of predictions between Mistral-8B and itself, $d_{\text{Conv}}$ denotes the KL-divergence of predictions of conventional ML algorithms and themselves, and $d_{\text{LLM}\to\text{method}}$ denotes the KL-divergence of predictions between Mistral-8B and conventional ML methods, where $\text{method} \in \{\text{DT}, \text{kNN}, \text{MLP}, \text{SVM}\}$.

| Case | $\bar{d}_{\text{LLM}}$ | $\bar{d}_{\text{Conv}}$ | $\bar{d}_{\text{LLM}\to\text{DT}}$ | $\bar{d}_{\text{LLM}\to\text{kNN}}$ | $\bar{d}_{\text{LLM}\to\text{MLP}}$ | $\bar{d}_{\text{LLM}\to\text{SVM}}$ |
|---|---|---|---|---|---|---|
| Standard (LLM) | 6.35 | - | 6.65 | 6.63 | 6.61 | 6.61 |
| Decision Tree (LLM) | 6.58 | 7.54 | 6.89 | 6.87 | 6.85 | 6.85 |
| $k$-NN (LLM) | 6.76 | 7.72 | 7.07 | 7.07 | 7.04 | 7.04 |
| MLP (LLM) | 6.15 | 7.53 | 6.49 | 6.47 | 6.44 | 6.45 |
| SVM (LLM) | 6.50 | 7.55 | 6.82 | 6.82 | 6.80 | 6.79 |

Table 10: **The KL-divergence results of predictions on the circle classification task, respectively, between Mistral-8B and itself, conventional machine learning algorithms and themselves, and Mistral-8B and conventional machine learning algorithms.** $d_{\text{LLM}}$ denotes the KL-divergence of predictions between Mistral-8B and itself, $d_{\text{Conv}}$ denotes the KL-divergence of predictions of conventional ML algorithms and themselves, and $d_{\text{LLM}\to\text{method}}$ denotes the KL-divergence of predictions between Mistral-8B and conventional ML methods, where $\text{method} \in \{\text{DT}, \text{kNN}, \text{MLP}, \text{SVM}\}$.

| Case | $\bar{d}_{\text{LLM}}$ | $\bar{d}_{\text{Conv}}$ | $\bar{d}_{\text{LLM}\to\text{DT}}$ | $\bar{d}_{\text{LLM}\to\text{kNN}}$ | $\bar{d}_{\text{LLM}\to\text{MLP}}$ | $\bar{d}_{\text{LLM}\to\text{SVM}}$ |
|---|---|---|---|---|---|---|
| Standard (LLM) | 5.81 | - | 6.07 | 6.03 | 6.05 | 6.04 |
| Decision Tree (LLM) | 5.63 | 7.64 | 5.95 | 5.92 | 5.93 | 5.93 |
| $k$-NN (LLM) | 5.42 | 7.33 | 5.75 | 5.72 | 5.73 | 5.73 |
| MLP (LLM) | 5.41 | 7.54 | 5.75 | 5.72 | 5.72 | 5.72 |
| SVM (LLM) | 5.54 | 7.56 | 5.84 | 5.81 | 5.82 | 5.82 |

Table 11: **The KL-divergence results of predictions on the moon classification task, respectively, between Mistral-8B and itself, conventional machine learning algorithms and themselves, and Mistral-8B and conventional machine learning algorithms.** $d_{\mathrm{LLM}}$ denotes the KL-divergence of predictions between Mistral-8B and itself, $d_{\mathrm{Conv}}$ denotes the KL-divergence of predictions of conventional ML algorithms and themselves, and $d_{\mathrm{LLM}\to\mathrm{method}}$ denotes the KL-divergence of predictions between Mistral-8B and conventional ML methods, where $\mathrm{method} \in \{\mathrm{DT}, \mathrm{kNN}, \mathrm{MLP}, \mathrm{SVM}\}$.

| Case | $\bar{d}_{\mathrm{LLM}}$ | $\bar{d}_{\mathrm{Conv}}$ | $\bar{d}_{\mathrm{LLM}\to\mathrm{DT}}$ | $\bar{d}_{\mathrm{LLM}\to\mathrm{kNN}}$ | $\bar{d}_{\mathrm{LLM}\to\mathrm{MLP}}$ | $\bar{d}_{\mathrm{LLM}\to\mathrm{SVM}}$ |
|---|---|---|---|---|---|---|
| Standard (LLM) | 7.39 | - | 7.59 | 7.59 | 7.56 | 7.56 |
| Decision Tree (LLM) | 7.39 | 7.80 | 7.62 | 7.60 | 7.58 | 7.58 |
| $k$-NN (LLM) | 7.50 | 7.61 | 7.77 | 7.76 | 7.73 | 7.74 |
| MLP (LLM) | 7.29 | 7.61 | 7.53 | 7.52 | 7.50 | 7.50 |
| SVM (LLM) | 7.41 | 7.59 | 7.66 | 7.65 | 7.62 | 7.63 |

Table 12: **The KL-divergence results of predictions on the linear classification task, respectively, between Qwen-2.5-7B and itself, conventional machine learning algorithms and themselves, and Qwen-2.5-7B and conventional machine learning algorithms.** $d_{\mathrm{LLM}}$ denotes the KL-divergence of predictions between Qwen-2.5-7B and itself, $d_{\mathrm{Conv}}$ denotes the KL-divergence of predictions of conventional ML algorithms and themselves, and $d_{\mathrm{LLM}\to\mathrm{method}}$ denotes the KL-divergence of predictions between Qwen-2.5-7B and conventional ML methods, where $\mathrm{method} \in \{\mathrm{DT}, \mathrm{kNN}, \mathrm{MLP}, \mathrm{SVM}\}$.

| Case | $\bar{d}_{\mathrm{LLM}}$ | $\bar{d}_{\mathrm{Conv}}$ | $\bar{d}_{\mathrm{LLM}\to\mathrm{DT}}$ | $\bar{d}_{\mathrm{LLM}\to\mathrm{kNN}}$ | $\bar{d}_{\mathrm{LLM}\to\mathrm{MLP}}$ | $\bar{d}_{\mathrm{LLM}\to\mathrm{SVM}}$ |
|---|---|---|---|---|---|---|
| Standard (LLM) | 8.31 | - | 8.50 | 8.46 | 8.46 | 8.45 |
| Decision Tree (LLM) | 7.97 | 7.54 | 8.15 | 8.10 | 8.10 | 8.10 |
| $k$-NN (LLM) | 8.13 | 7.72 | 8.33 | 8.27 | 8.27 | 8.27 |
| MLP (LLM) | 8.03 | 7.55 | 8.24 | 8.19 | 8.19 | 8.19 |
| SVM (LLM) | 7.97 | 7.53 | 8.17 | 8.13 | 8.13 | 8.12 |

Table 13: **The KL-divergence results of predictions on the circle classification task, respectively, between Qwen-2.5-7B and itself, conventional machine learning algorithms and themselves, and Qwen-2.5-7B and conventional machine learning algorithms.** $d_{\mathrm{LLM}}$ denotes the KL-divergence of predictions between Qwen-2.5-7B and itself, $d_{\mathrm{Conv}}$ denotes the KL-divergence of predictions of conventional ML algorithms and themselves, and $d_{\mathrm{LLM}\to\mathrm{method}}$ denotes the KL-divergence of predictions between Qwen-2.5-7B and conventional ML methods, where $\mathrm{method} \in \{\mathrm{DT}, \mathrm{kNN}, \mathrm{MLP}, \mathrm{SVM}\}$.

| Case | $\bar{d}_{\mathrm{LLM}}$ | $\bar{d}_{\mathrm{Conv}}$ | $\bar{d}_{\mathrm{LLM}\to\mathrm{DT}}$ | $\bar{d}_{\mathrm{LLM}\to\mathrm{kNN}}$ | $\bar{d}_{\mathrm{LLM}\to\mathrm{MLP}}$ | $\bar{d}_{\mathrm{LLM}\to\mathrm{SVM}}$ |
|---|---|---|---|---|---|---|
| Standard (LLM) | 8.49 | - | 8.70 | 8.70 | 8.70 | 8.70 |
| Decision Tree (LLM) | 8.52 | 7.64 | 8.76 | 8.74 | 8.75 | 8.75 |
| $k$-NN (LLM) | 8.72 | 7.33 | 8.91 | 8.90 | 8.90 | 8.90 |
| MLP (LLM) | 8.69 | 7.56 | 8.90 | 8.89 | 8.89 | 8.90 |
| SVM (LLM) | 8.35 | 7.54 | 8.58 | 8.56 | 8.56 | 8.56 |

Table 14: **The KL-divergence results of predictions on the moon classification task, respectively, between Qwen-2.5-7B and itself, conventional machine learning algorithms and themselves, and Qwen-2.5-7B and conventional machine learning algorithms.** $d_{\mathrm{LLM}}$ denotes the KL-divergence of predictions between Qwen-2.5-7B and itself, $d_{\mathrm{Conv}}$ denotes the KL-divergence of predictions of conventional ML algorthms and themselves, and $d_{\mathrm{LLM}\to\mathrm{method}}$ denotes the KL-divergence of predictions between Qwen-2.5-7B and conventional ML methods, where $\mathrm{method} \in \{\mathrm{DT}, \mathrm{kNN}, \mathrm{MLP}, \mathrm{SVM}\}$.

| Case | $\bar{d}_{\mathrm{LLM}}$ | $\bar{d}_{\mathrm{Conv}}$ | $\bar{d}_{\mathrm{LLM}\to\mathrm{DT}}$ | $\bar{d}_{\mathrm{LLM}\to\mathrm{kNN}}$ | $\bar{d}_{\mathrm{LLM}\to\mathrm{MLP}}$ | $\bar{d}_{\mathrm{LLM}\to\mathrm{SVM}}$ |
|---|---|---|---|---|---|---|
| Standard (LLM) | 8.02 | - | 8.28 | 8.25 | 8.25 | 8.25 |
| Decision Tree (LLM) | 8.65 | 7.80 | 8.90 | 8.87 | 8.88 | 8.87 |
| $k$-NN (LLM) | 9.03 | 7.61 | 9.30 | 9.27 | 9.27 | 9.27 |
| MLP (LLM) | 8.79 | 7.59 | 9.04 | 9.02 | 9.03 | 9.01 |
| SVM (LLM) | 8.33 | 7.61 | 8.59 | 8.56 | 8.56 | 8.56 |

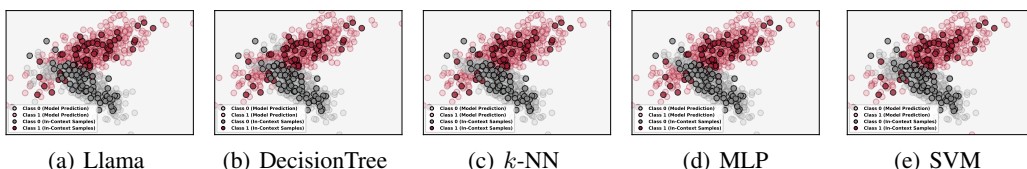

(a) Llama (b) DecisionTree (c) $k$-NN (d) MLP (e) SVM

Figure 60: **Visualizations of predictions of Mistral-8B and conventional ML methods in the linear classification task that only contains in-distribution query data**. Fig. (a) visualizes the predictions of Mistral-8B. Figs. (b)-(e) visualize the predictions of conventional machine learning methods. According to the visualizations, we can observe that the behavior of Mistral-8B resembles that of conventional machine learning methods. This phenomenon matches the quantitative results reported in Tables 9.

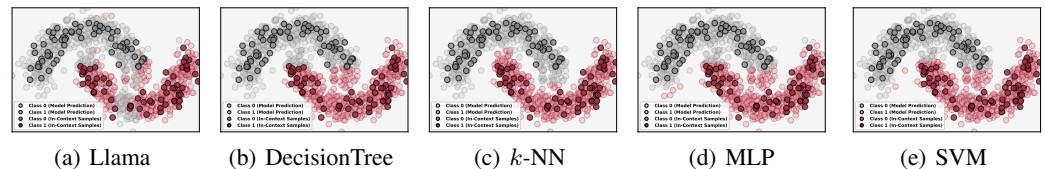

(a) Llama (b) DecisionTree (c) $k$-NN (d) MLP (e) SVM

Figure 61: **Visualizations of predictions of Mistral-8B and conventional ML methods in the moon classification task that only contains in-distribution query data**. Fig. (a) visualizes the predictions of Mistral-8B. Figs. (b)-(e) visualize the predictions of conventional machine learning methods. According to the visualizations, we can observe that the behavior of Mistral-8B resembles that of conventional machine learning methods. This phenomenon matches the quantitative results reported in Tables 11.

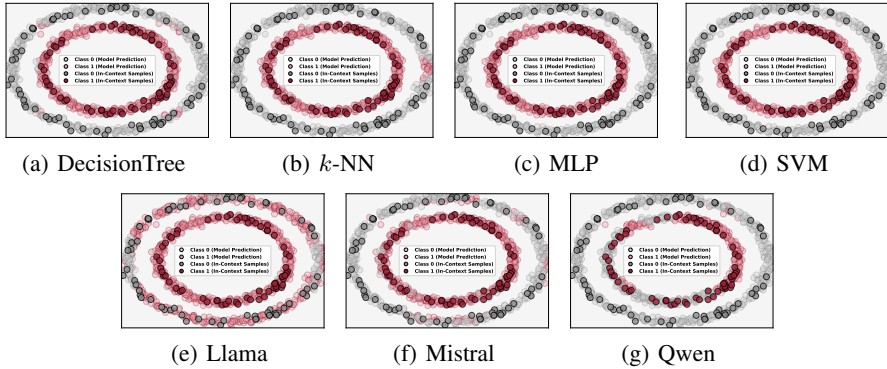

(a) DecisionTree (b) $k$-NN (c) MLP (d) SVM

(e) Llama (f) Mistral (g) Qwen

Figure 62: **Visualizations of predictions of Mistral-8B and conventional ML methods in the circle classification task that only contains in-distribution data**. Figs. (a)-(d) visualizes the predictions of conventional ML methods. Figs. (e)-(g) visualize the predictions of Llama-3-8B, Mistral-8B, and Qwen-2.5-7B.

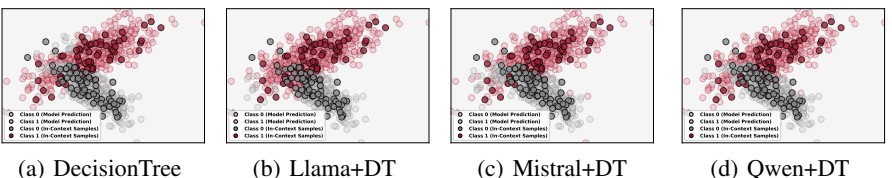

(a) DecisionTree (b) Llama+DT (c) Mistral+DT (d) Qwen+DT

Figure 63: **Visualizations of predictions of LLMs prompted with prompts where Decision Tree is specified and conventional Decision Tree methods in the linear classification task that only contains in-distribution data**. Fig. (a) visualizes the predictions of conventional Decision Tree methods. Figs. (b)-(d) visualize the predictions of Llama-3-8B, Mistral-8B, and Qwen-2.5-7B with prompts where Decision Tree is specified.

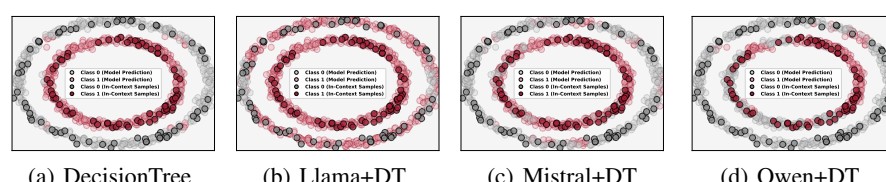

(a) DecisionTree   (b) Llama+DT   (c) Mistral+DT   (d) Qwen+DT

Figure 64: **Visualizations of predictions of LLMs prompted with prompts where Decision Tree is specified and conventional Decision Tree in the circle classification task that only contains in-distribution data**. Fig. (a) visualizes the predictions of conventional Decision Tree methods. Figs. (b)-(d) visualize the predictions of Llama-3-8B, Mistral-8B, and Qwen-2.5-7B with prompts where Decision Tree is specified.

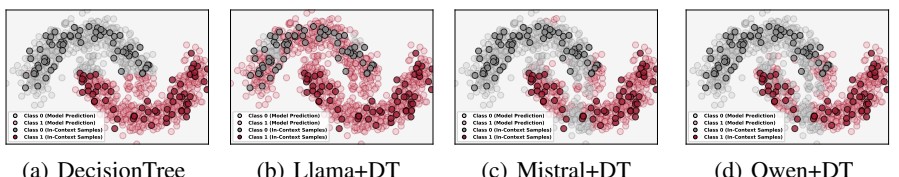

(a) DecisionTree   (b) Llama+DT   (c) Mistral+DT   (d) Qwen+DT

Figure 65: **Visualizations of predictions of LLMs prompted with prompts where Decision Tree is specified and conventional Decision Tree in the moon classification task that only contains in-distribution data**. Fig. (a) visualizes the predictions of conventional Decision Tree methods. Figs. (b)-(d) visualize the predictions of Llama-3-8B, Mistral-8B, and Qwen-2.5-7B with prompts where Decision Tree is specified.

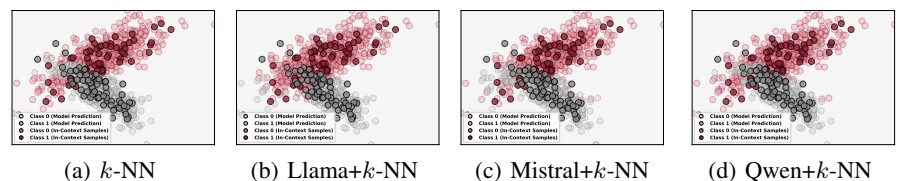

(a) $k$-NN   (b) Llama+$k$-NN   (c) Mistral+$k$-NN   (d) Qwen+$k$-NN

Figure 66: **Visualizations of predictions of LLMs prompted with prompts where $k$-NN is specified and conventional $k$-NN methods in the linear classification task that only contains in-distribution data**. Fig. (a) visualizes the predictions of conventional $k$-NN methods. Figs. (b)-(d) visualize the predictions of Llama-3-8B, Mistral-8B, and Qwen-2.5-7B with prompts where $k$-NN is specified.

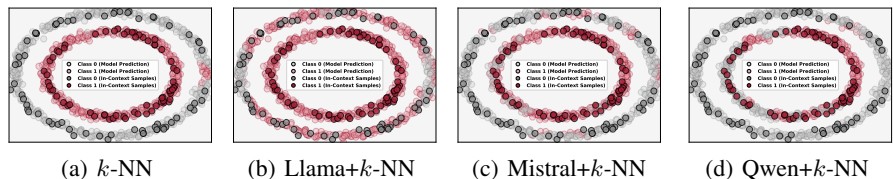

(a) $k$-NN   (b) Llama+$k$-NN   (c) Mistral+$k$-NN   (d) Qwen+$k$-NN

Figure 67: **Visualizations of predictions of LLMs prompted with prompts where $k$-NN is specified and conventional $k$-NN in the circle classification task that only contains in-distribution data**. Fig. (a) visualizes the predictions of conventional $k$-NN methods. Figs. (b)-(d) visualize the predictions of Llama-3-8B, Mistral-8B, and Qwen-2.5-7B with prompts where $k$-NN is specified.

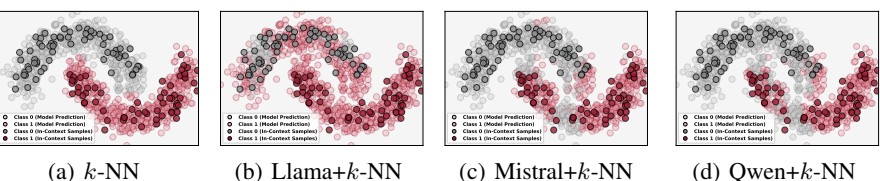

(a) $k$-NN   (b) Llama+$k$-NN   (c) Mistral+$k$-NN   (d) Qwen+$k$-NN

Figure 68: **Visualizations of predictions of LLMs prompted with prompts where $k$-NN is specified and conventional $k$-NN in the moon classification task that only contains in-distribution data**. Fig. (a) visualizes the predictions of conventional $k$-NN methods. Figs. (b)-(d) visualize the predictions of Llama-3-8B, Mistral-8B, and Qwen-2.5-7B with prompts where $k$-NN is specified.

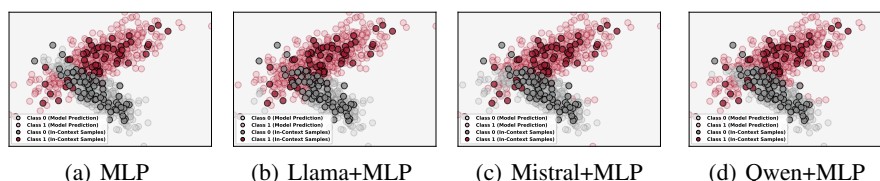

|(a) MLP|(b) Llama+MLP|(c) Mistral+MLP|(d) Qwen+MLP|

Figure 69: **Visualizations of predictions of LLMs prompted with prompts where MLP is specified and conventional MLP methods in the linear classification task that only contains in-distribution data**. Fig. (a) visualizes the predictions of conventional MLP methods. Figs. (b)-(d) visualize the predictions of Llama-3-8B, Mistral-8B, and Qwen-2.5-7B with prompts where MLP is specified.

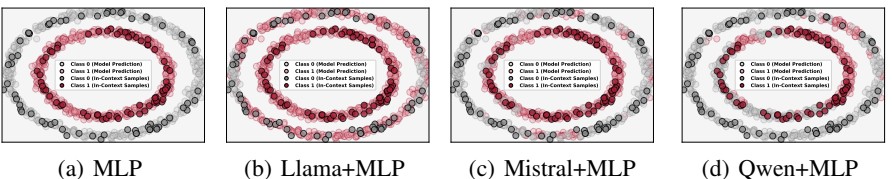

|(a) MLP|(b) Llama+MLP|(c) Mistral+MLP|(d) Qwen+MLP|

Figure 70: **Visualizations of predictions of LLMs prompted with prompts where MLP is specified and conventional MLP in the circle classification task that only contains in-distribution data**. Fig. (a) visualizes the predictions of conventional MLP methods. Figs. (b)-(d) visualize the predictions of Llama-3-8B, Mistral-8B, and Qwen-2.5-7B with prompts where MLP is specified.

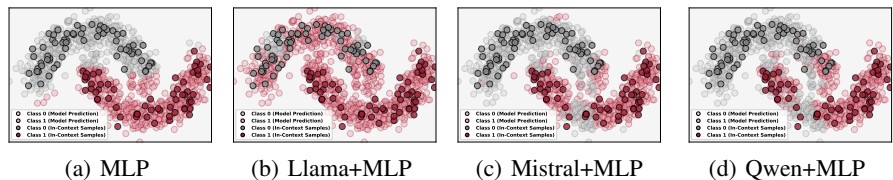

|(a) MLP|(b) Llama+MLP|(c) Mistral+MLP|(d) Qwen+MLP|

Figure 71: **Visualizations of predictions of LLMs prompted with prompts where MLP is specified and conventional MLP in the moon classification task that only contains in-distribution data**. Fig. (a) visualizes the predictions of conventional MLP methods. Figs. (b)-(d) visualize the predictions of Llama-3-8B, Mistral-8B, and Qwen-2.5-7B with prompts where MLP is specified.

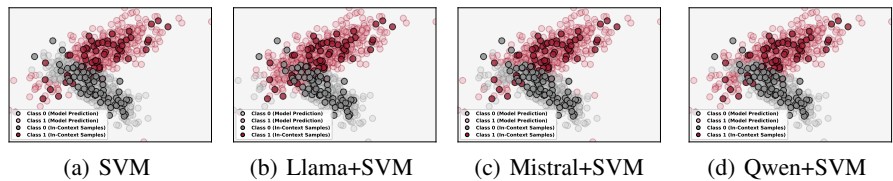

|(a) SVM|(b) Llama+SVM|(c) Mistral+SVM|(d) Qwen+SVM|

Figure 72: **Visualizations of predictions of LLMs prompted with prompts where SVM is specified and conventional SVM methods in the linear classification task that only contains in-distribution data**. Fig. (a) visualizes the predictions of conventional SVM methods. Figs. (b)-(d) visualize the predictions of Llama-3-8B, Mistral-8B, and Qwen-2.5-7B with prompts where SVM is specified.

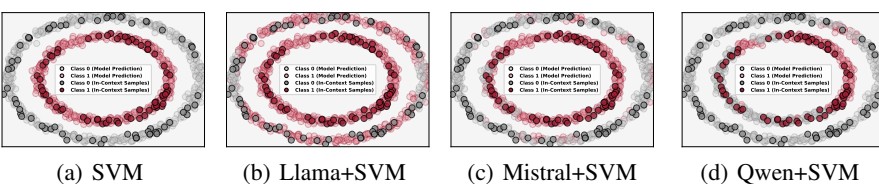

|(a) SVM|(b) Llama+SVM|(c) Mistral+SVM|(d) Qwen+SVM|

Figure 73: **Visualizations of predictions of LLMs prompted with prompts where SVM is specified and conventional SVM in the circle classification task that only contains in-distribution data**. Fig. (a) visualizes the predictions of conventional SVM methods. Figs. (b)-(d) visualize the predictions of Llama-3-8B, Mistral-8B, and Qwen-2.5-7B with prompts where SVM is specified.

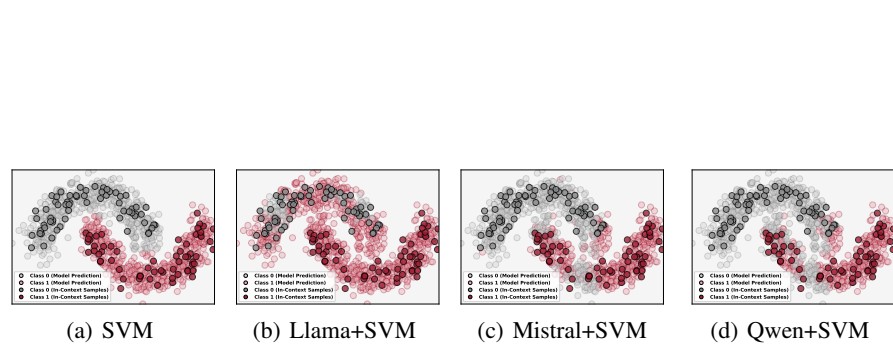

(a) SVM      (b) Llama+SVM      (c) Mistral+SVM      (d) Qwen+SVM

Figure 74: **Visualizations of predictions of LLMs prompted with prompts where SVM is specified and conventional SVM in the moon classification task that only contains in-distribution data.** Fig. (a) visualizes the predictions of conventional SVM methods. Figs. (b)-(d) visualize the predictions of Llama-3-8B, Mistral-8B, and Qwen-2.5-7B with prompts where SVM is specified.

## G DETAILS OF PROMPTS

In this paper, in order to probe the behavior of LLMs in classification tasks with in-context learning, we have LLMs perform classification tasks with both implicit and explicit reasoning paradigms. These cases are treated as the standard cases. The corresponding prompt for the standard reasoning is:

---

**A Prompt Example for Standard Implicit & Explicit Reasoning**

### Instruction: Given pairs of numbers and their labels, predict the label for a new input pair of numbers based on the provided data. Answer with only one of the labels 0 and 1, and evaluate your confidence for the answer with a float number between 0.0 and 1.0, where 0.0 means you are absolutely inconfident to your answer while 1.0 means you are absolutely confident to your answer. **(IR:)** Please do not provide any reasoning process in your response. / **(ER:)** Please provide detailed reasoning process in your response. Let's think step by step.
### Input:
Input: 2.327369299801017 2.238478737209186
Label: 1
Input: -0.7246972544778265 0.3996389489449079
Label: 0
...
Input: 2.2105720569686538 0.9862902654079408
Label: 1
What is the label for this input? And how confident you are to your answer?
Input: -3.915171090551515 -1.1395254205266334
Label:
### Response:   Label:   Confidence:   **(ER:)** Reasoning Process:

---

According to the results (i.e., the outputs of explicit reasoning), we notice that LLMs tend to address the classification task in the way of machine learning methods. This inspires us to determine whether specified machine learning methods contribute to improving the decision boundaries of LLMs. Thus, we further propose to conduct experiments to explore the effect of specified machine learning methods on classification tasks. This goal is achieved by slightly modifying the prompts adopted in the standard implicit and explicit reasoning. Specifically, compared to the prompts of the standard reasoning paradigms, the prompts for reasoning with specified machine learning methods are modified with the instruction to have LLMs perform inference with specified machine learning algorithms. To make it clear, an example of prompts with specified machine learning methods is presented in the following.

---

**A Prompt Example for Implicit & Explicit Reasoning with Specified Methods**

### Instruction: Given pairs of numbers and their labels, please apply **{*machine learning*}** method(s) to predict the label for a new input pair of numbers based on the provided data. Answer with only one of the labels 0 and 1, and evaluate your confidence for the answer with a float number between 0.0 and 1.0, where 0.0 means you are absolutely inconfident to your answer while 1.0 means you are absolutely confident to your answer. **(IR:)** Please do not provide any reasoning process in your response. / **(ER:)** Please provide detailed reasoning process in your response. Let's think step by step.
### Input:
Input: 2.327369299801017 2.238478737209186
Label: 1
Input: -0.7246972544778265 0.3996389489449079
Label: 0
...
Input: 2.2105720569686538 0.9862902654079408
Label: 1
What is the label for this input? And how confident you are to your answer?
Input: -3.915171090551515 -1.1395254205266334
Label:
### Response:   Label:   Confidence:   **(ER:)** Reasoning Process:

---

