# OpenReview forum: "A Closer Look at In-context Learning of LLMs in Simple Classification Tasks"
_ICLR.cc/2026/Conference — Submitted to ICLR 2026_

### Official Review · Reviewer_ev9p · 2025-10-26

**Soundness:** 4
**Presentation:** 3
**Contribution:** 2
**Rating:** 4
**Confidence:** 4

**Summary:**

This paper investigates the in-context learning (ICL) behavior of large language models (LLMs) on simple classification tasks, focusing on their decision boundaries and reasoning paradigms (implicit vs. explicit). The authors introduce a quantitative metric—Smooth Score—to measure the fragmentation of decision boundaries and analyze whether LLMs implicitly follow traditional machine learning (ML) algorithms. The study provides interesting empirical insights into how LLMs behave like conventional ML classifiers in simple discriminative settings.

**Strengths:**

1. Clear motivation and relevance. The work addresses an important question in understanding the mechanisms underlying ICL, which is highly relevant to the ICLR community.

2. Novel quantitative perspective. The introduction of the Smooth Score offers a new quantitative handle to study decision boundary smoothness, extending prior qualitative analyses.

3. Systematic empirical setup. The authors compare multiple reasoning modes (implicit vs. explicit) and multiple LLMs (Llama, Qwen, DeepSeek, etc.), which increases the robustness of the observations.

4. Interesting findings. The observation that implicit reasoning yields smoother decision boundaries than explicit reasoning provides a counterintuitive and potentially valuable insight into the interference effect of CoT-like prompting.

5. Comprehensive experiments, exquisite figures and layout.

**Weaknesses:**

1. Insufficient discussion of implications. The work stops short of discussing how these findings could inform model design or prompting strategies for real-world tasks. For instance, what is the significance of understanding or improving the non-smoothness in this LLM classification process for real-world classification tasks?
2. Shallow theoretical contribution. Although the Smooth Score and KL-based analyses are well-defined, the paper mainly remains empirical and descriptive without deeper theoretical justification for why implicit reasoning aligns better with ML-like boundaries.
3. Limited novelty . The study is largely built upon the framework proposed by Zhao et al., and compared with their work, it does not provide deeper insights or more compelling conclusions.
4. Some formatting issues. Several sections (e.g., Figure captions, equation formatting, and cross-references like “Section ??”) need editorial polishing.

**Questions:**

1. How do the observations transfer to textual or multimodal classification tasks beyond numeric toy data?
2. What are the clear innovations compared to the work of Zhao et al.?

---

> ### Author Response · Authors · 2025-11-27
>
> > W1. 1. Insufficient discussion of implications. The work stops short of discussing how these findings could inform model design or prompting strategies for real-world tasks. For instance, what is the significance of understanding or improving the non-smoothness in this LLM classification process for real-world classification tasks?
>
> **Response:** Thank you for your comments. According to your comments, we would like to respectfully argue that it is significant to understanding the behaviors of LLMs on the conventional ML classification task. Moreover, we have provided two potential solutions in our paper.
>
> **_[Significant.]_** Compared to conventional reasoning tasks (e.g., math reasoning), the simple classification is a perfect benchmark to probe the true in-context learning capability of LLMs. Specifically, the core insight of in-context learning is understanding the patterns behind the given in-context samples and performing tasks accordingly. Different from conventional reasoning tasks, the in-context samples in ML classification tasks are a series of data with coordinates without concrete high-level semantic information. Thus, in this case, LLMs have to infer the patterns directly from the two sets of data and predict corresponding labels, which is harder than conventional text-based reasoning tasks.
>
> **_[Inspirations of findings.]_** In our work, we show that LLMs can roughly handle the conventional ML classification tasks, which demonstrates the powerful in-context capability of LLMs. Two inspirations are worth attention here. On the one hand, LLMs with implicit reasoning perform better on these tasks and behave similarly to the conventional ML methods. On the other hand, the poor performance of explicit reasoning is mainly derived from high perplexity. According to our empirical results, the poor performance can be alleviated by voting answers from multiple reasoning trajectories. Thus, for similar tasks, we can either select implicit reasoning or perform majority voting among multiple reasoning trajectories.
>
> > W2. 1. Shallow theoretical contribution. Although the Smooth Score and KL-based analyses are well-defined, the paper mainly remains empirical and descriptive without deeper theoretical justification for why implicit reasoning aligns better with ML-like boundaries.
>
> **Response:** For your concern about intuitive explanations, we would like to first clarify that **the simple ML classification task is a perfect synthetic benchmark to evaluate the true in-context capability of LLMs**. The core insight of in-context learning is identifying the patterns behind the given in-context samples and predicting the labels of query data accordingly. In the simple ML classification task, the in-context data are a set of coordinates without any high-level semantic information. Thus, LLMs have to explore the patterns of the two sets of data to further predict the query data without any assistance from the knowledge of the world.

---

> ### Author Response · Authors · 2025-11-27
>
> > W3. 1. Limited novelty . The study is largely built upon the framework proposed by Zhao et al., and compared with their work, it does not provide deeper insights or more compelling conclusions.
> > > Q2. 1. What are the clear innovations compared to the work of Zhao et al.?
>
> **Response:** Thank you for your comments. According to your comments, your main concern mainly focuses on the novelty of our paper. Here, **we would like to respectfully argue that our work is NOT a simple consequent work of Zhao et al., although our work is built upon its problem**.
>
> Zhao et al. is the first work that proposes to evaluate the in-context capability of LLMs on conventional machine learning classification tasks. In this work, they found that LLMs perform poorly on these tasks and obtain fragmented decision boundaries. Then, they investigate several experimental settings, ranging from the size of models and in-context data to the order of in-context samples and the format of labels, to explore these phenomena. Finally, they found that only fine-tuning can help mitigate the fragmented decision boundaries. **In general, Zhao et al. proposes a new evaluation benchmark and initially explores the in-context capability of LLMs on this benchmark.**
>
> However, in our work, **we notice that the synthetic benchmark is a hard yet effective benchmark to evaluate the real reasoning capability of LLMs**. Specifically, compared to text-based reasoning tasks, the machine learning classification task hardly contains any high-level semantic information. Meanwhile, since the query data are directly derived from the plane/space where the in-context samples lie, there isn't any label information, and the query data include both in-distribution (ID) and out-of-distribution (OOD) data. Thus, LLMs have to summarize the patterns from the given in-context samples and perform reasoning on both ID and OOD query data. This imposes challenges to the reasoning and generalization capabilities of LLMs. **Based on these observations, we take a further step to understand the behaviors of LLMs in in-context learning.** Our innovations can be summarized as follows:
> - **We investigate the behaviors of LLMs in in-context learning with both implicit and explicit reasoning**, while Zhao et al. only performed reasoning only with implicit reasoning. In our work, **we find that explicit reasoning unexpectedly achieves worse performance than implicit reasoning, which is different from the common sense that explicit reasoning outperforms implicit reasoning in conventional reasoning tasks**.
> - **We further probe the confidence of LLMs to investigate the behaviors of LLMs when they are performing in-context learning**. To ensure that the investigation is comprehensive, we probe both subjective and objective confidence of LLMs. According to the results, we find that **the subjective and objective evaluations of LLMs for the reasoning are completely different, which indicates the inconsistency of the expression and generation of reasoning**.
> - With the high perplexity, we conjecture that the fragmented decision boundaries may be derived from uncertain reasoning processes. Thus, we propose to adopt SC-CoT to mitigate the randomness resulting from the uncertainty. According to our empirical results, we find that the SC-CoT helps alleviate the fragmented decision boundaries.
> - Motivated by the observation that LLMs with implicit reasoning obtain similar decision boundaries to those derived from conventional ML methods, we measure the distributions of predictions of both LLMs and conventional ML methods via KL-divergence. The quantitative results show that the distributions of the two sets of predictions are highly similar, implying that LLMs with implicit reasoning perform in a similar way to conventional machine learning methods.

---

> ### Author Response · Authors · 2025-11-27
>
> > W4. 1. Some formatting issues. Several sections (e.g., Figure captions, equation formatting, and cross-references like “Section ??”) need editorial polishing.
>
> **Response:** Thank you for your careful review. We will modify these typos in the updated version of our paper.
>
> > Q1. How do the observations transfer to textual or multimodal classification tasks beyond numeric toy data?
>
> **Answer:** Thank you for the question. According to your question, we rephrase your textual classification tasks as tasks like sentiment analysis. As far as we know, language models, such as RoBERTa, have already achieved about 95% accuracy on sentiment classification tasks. In the context of LLMs, Llama-3.1-405B can achieve about 94% accuracy under zero-shot reasoning settings. For the multimodal classification tasks (e.g., VQA), Qwen2-VL-7B can achieve about 88% accuracy on the VQAv2 benchmark, and Llama3.2-11B achieves 85% accuracy.
>
> Here, we would like to clarify that the simple classification tasks adopted in our paper are different from the tasks mentioned above.
> - On the one hand, the simple machine learning classification task is not a typical NLP task. In our work, we transfer the conventional ML problem into a reasoning problem by describing data and tasks with language instead of matrices/vectors.  In such a task, the LLMs have to summarize patterns from the data since there isn't any world knowledge available. This imposes a severe challenge to the reasoning capability of LLMs
> - On the other hand, the simple ML classification task helps probe the low-level capability (e.g., numerical calculation) of LLMs, while the tasks mentioned in your question mainly focus on the high-level reasoning capability.
> - Moreover, since the query data are directly derived from the plane/space where the in-context data lie, the generated data lack labels and include both in-distribution and out-of-distribution data. This further imposes a challenge to the generalization performance of LLMs.

---

### Official Review · Reviewer_Tr9z · 2025-10-31

**Soundness:** 4
**Presentation:** 2
**Contribution:** 2
**Rating:** 4
**Confidence:** 4

**Summary:**

This paper investigates how large language models (LLMs) perform in-context learning (ICL) in classification tasks. The authors observe that LLMs often exhibit irregular and fragmented decision boundaries, where implicit reasoning tends to produce smoother and more consistent boundaries than explicit reasoning. Their analysis suggests that, during discriminative tasks, LLMs implicitly emulate traditional machine learning algorithms, revealing a connection between contextual token-based reasoning and algorithmic behavior.

**Strengths:**

1. The paper provides extensive experimental evidence demonstrating the existence of irregular decision boundaries in LLMs during ICL.
2. The exploration of fragmented decision boundaries may offers valuable insight into the inner mechanisms of ICL, contributing to a deeper understanding of one of the most central capabilities of current LLMs.

**Weaknesses:**

1. The findings rely heavily on empirical observations, with limited theoretical analysis or intuitive explanation of why such irregularities emerge.
2. The writing quality is poor—the abstract and introduction mainly describe what was done, without clearly articulating the key findings, their implications, or how they advance understanding within the community.
3. The experiments are restricted to models under 8B parameters, leaving uncertainty about whether the observed fragmentation persists at larger scales. Consequently, the scalability and practical implications of the conclusions remain unclear.

**Questions:**

How might different forms of explicit reasoning (e.g., Tree of Thoughts, self-critique, or reflective reasoning frameworks) influence the resulting decision boundaries? Would such reasoning strategies smooth out or exacerbate the irregularities observed compared to standard Chain of Thought reasoning?

---

> ### Author Response · Authors · 2025-11-27
>
> > W1. The findings rely heavily on empirical observations, with limited theoretical analysis or intuitive explanation of why such irregularities emerge.
>
> **Response:** Thank you for your comments.
>
> For your concern about theory, we agree that theoretical analysis helps better understand the behavior of LLMs. However, a prerequisite for theoretical analysis is a sound theoretical framework. Due to the complexity of LLMs, such a framework is lacking. Thus, we cannot provide sufficient theoretical analysis here. According to your concern, we will list this as future work in our paper.
>
> For your concern about intuitive explanations, we would like to first clarify that **the simple ML classification task is a perfect synthetic benchmark to evaluate the true in-context capability of LLMs**. The core insight of in-context learning is identifying the patterns behind the given in-context samples and predicting the labels of query data accordingly. In the simple ML classification task, the in-context data are a set of coordinates without any high-level semantic information. Thus, LLMs have to explore the patterns of the two sets of data to further predict the query data without any assistance from the knowledge of the world.
>
> Moreover, **for the phenomenon that LLMs with implicit reasoning perform better than those with explicit reasoning, we also provide some intuitive conjectures and empirical demonstrations**.
> - On the one hand, we notice that LLMs with explicit reasoning predict the labels of query data with high perplexity. **This phenomenon implies that the more fragmented decision boundaries in explicit reasoning result from the randomness derived from the uncertainty**. To demonstrate this, we further adopt SC-CoT to ease uncertainty by voting answers from multiple reasoning samples.
> - On the other hand, we observe that the decision boundaries generated from LLMs with implicit reasoning are smoother and resemble those derived from conventional machine learning methods. **This motivates us to demonstrate whether LLMs with implicit reasoning perform similarly to conventional ML methods**. Thus, in Section 5, we leverage KL-divergence to compare the distribution differences between the predictions of LLMs and conventional ML methods, demonstrating that LLMs with implicit reasoning act in a similar way to conventional ML methods.
>
> All these contributions in our paper demonstrate that we conducted reasonable and intuitive conjectures, demonstrations, and explanations of the irregularities.
>
> > W2. The writing quality is poor—the abstract and introduction mainly describe what was done, without clearly articulating the key findings, their implications, or how they advance understanding within the community.
>
> **Response:** Thank you for your comments. We agree that the initial version of the abstract did not sufficiently highlight the insights of our paper. **To address your concern, we will modify our abstract section in the updated version of our paper**. Instead of merely listing experimental results, the revised content will focus on key findings and the corresponding implications.
>
> > W3. The experiments are restricted to models under 8B parameters, leaving uncertainty about whether the observed fragmentation persists at larger scales. Consequently, the scalability and practical implications of the conclusions remain unclear.
>
> **Response:** Thank you for your comments. **Except for the 8B model, we also have conducted experiments on DeepSeek-V3 in our paper.** The results are reported in Appendix D3. According to our reported visualization results, we find that
> - In the context of implicit reasoning, consistent with small LLMs (e.g., Llama3-8B and Qwen2.5-7B), **the decision boundaries derived from DeepSeek-V3 are consistently fragmented**, and **specifying machine learning algorithms in prompts does not help generate better decision boundaries**.
> - In the context of implicit reasoning, consistent with small LLMs, **both subjective and objective confidence of DeepSeek-V3 are improved when machine learning methods are specified in prompts**. Specifically, the subjective confidence increases while the perplexity of predictions decreases.
> - In the context of explicit reasoning, consistent with small LLMs, **the decision boundaries derived from DeepSeek-V3 become even worse than those derived with implicit reasoning**, and **specifying machine learning algorithms in prompts does not help**.
> - In the context of explicit reasoning, consistent with small LLMs, **both subjective and objective confidence of DeepSeek-V3 are improved when machine learning methods are specified in prompts**.
> - Different from that in small LLMs, applying SC-CoT does not help improve the smoothness of decision boundaries.
>
> To further address your concerns, we conduct experiments on LLMs with about 70B parameters. We will merge these experiments in the updated version of our paper.

---

> ### Author Response · Authors · 2025-11-27
>
> > Q. How might different forms of explicit reasoning (e.g., Tree of Thoughts, self-critique, or reflective reasoning frameworks) influence the resulting decision boundaries? Would such reasoning strategies smooth out or exacerbate the irregularities observed compared to standard Chain of Thought reasoning?
>
> **Response:** Thank you for your comments. We think that your suggestions here are reasonable. However, **according to our observations in Table 1, the main reason for the fragmented decision boundaries is the incapability of low-level numerical execution and noisy emulation of ML algorithms**. Thus, from this perspective, although advanced explicit reasoning frameworks (e.g., ToT, self-critique) are powerful, they primarily aim to reduce factual or logical mistakes in long-horizon symbolic reasoning. They do not fundamentally improve the LLMs' capability in performing stable, high-precision numerical operations on hundreds of floating-point numbers. Thus, we conjecture that explicit reasoning tends to hurt the smoothness of decision boundaries because it forces the model to verbalize and execute fragile numerical steps, and more elaborate verbal reasoning schemes are unlikely to remove this fundamental bottleneck.

---

### Official Review · Reviewer_nLo7 · 2025-10-31

**Soundness:** 3
**Presentation:** 3
**Contribution:** 2
**Rating:** 4
**Confidence:** 3

**Summary:**

This paper investigates why Large Language Models (LLMs) produce unexpectedly fragmented decision boundaries when performing in-context learning (ICL) on simple machine learning classification tasks. It finds this is primarily due to a generalization failure caused by a distribution mismatch between the query data (OOD) and the in-context data.
The paper presents several in-depth analyses:
1. Using a statistical method based on KL-divergence, it demonstrates that under standard implicit reasoning (without specific instructions), the predictive behavior of LLMs is statistically "consistent with" the ideal behavior of traditional ML algorithms (e.g., k-NN, SVM, MLP). This suggests LLMs do intrinsically and implicitly leverage an effective classification logic.
2. The paper finds that the "uniform grid" query data, which caused fragmented boundaries in prior experiments, is Out-of-Distribution (OOD) relative to the "clustered" in-context examples seen by the LLM. When the query data is replaced with In-Distribution (ID) data (i.e., also sampled from the clusters), the LLM's classification performance significantly improves and aligns with traditional ML algorithms. This proves that LLMs are not incapable of classification, but rather that their ICL generalization to OOD data is poor.

**Strengths:**

1. The paper's experimental design is highly systematic. Starting from the phenomenon of "fragmented decision boundaries," it rigorously rules out various possibilities through carefully designed comparative experiments (e.g., implicit reasoning vs. explicit reasoning, standard prompts vs. specified ML algorithm prompts).
2. The paper does not stop at the superficial conclusion that "LLMs are bad at classification." Instead, it delves deeper to convincingly demonstrate for the first time that the root cause is OOD generalization failure—a distribution mismatch between the "clustered" examples (seen) and the "uniform grid" data (predicted).
3. The paper proposes two simple and novel tools: (1) the "Smooth Score," which for the first time quantitatively measures the fragmentedness of decision boundaries, moving analysis beyond subjective observation; and (2) a KL-divergence-based statistical method to determine if an LLM's behavior is "consistent with" traditional ML algorithms, offering a new approach for black-box model analysis.

**Weaknesses:**

1. The analysis is primarily based on 8B-level models (Llama-3, Mistral, Qwen) and one DeepSeek-V3. It lacks systematic verification across a wider range of model scales (e.g., 70B+ or 3B-), making it uncertain if the conclusions apply to all LLMs.
2. All experiments are conducted almost exclusively on 2D (and one 3D) synthetic data. The paper's key metric, the "Smooth Score," is also acknowledged to be unscalable to high dimensions. Therefore, it is unknown if these findings will hold for real-world, high-dimensional, multi-class classification tasks.

**Questions:**

1. How is the validity of the "Smooth Score" verified? If a model (LLM) outputs a "trivial solution," such as predicting the same class (e.g., all 0s) for all query points, its "Smooth Score" would be a perfect 1.0.
2. The "KL-divergence-based statistical method" should report variance (or standard deviations) for its averaged results, in order to demonstrate whether the "gaps" (which the paper relies on for comparison) are statistically significant.

---

> ### Author Response · Authors · 2025-11-27
>
> > W1. 1. The analysis is primarily based on 8B-level models (Llama-3, Mistral, Qwen) and one DeepSeek-V3. It lacks systematic verification across a wider range of model scales (e.g., 70B+ or 3B-), making it uncertain if the conclusions apply to all LLMs.
>
> **Response:** Thanks for your comments. According to your comments, we notice that your main concern here is that the phenomena observed in our paper are not validated on other sizes of LLMs. To address your concern, we conduct the same set of experiments on both Qwen2.5-72B and Llama-3-70B. Due to the large size of the models and the large number of tasks, the experiments are still running; we will provide results in the updated version of our paper.
>
> > W2. 1. All experiments are conducted almost exclusively on 2D (and one 3D) synthetic data. The paper's key metric, the "Smooth Score," is also acknowledged to be unscalable to high dimensions. Therefore, it is unknown if these findings will hold for real-world, high-dimensional, multi-class classification tasks.
>
> **Response:** Thanks for your comments. For your concern regarding the generalization of our works in real-world, high-dimensional, and multi-class classification tasks, we provide clarifications as follows.
>
> **_[Concern about Dataset.]_** Compared to conventional reasoning tasks, such as math reasoning, **performing reasoning on these classification tasks helps evaluate the true reasoning capability of LLMs**. Specifically, in our paper, we transfer the conventional ML classification tasks to reasoning tasks by describing the tasks with texts instead of matrices or vectors. In such reasoning tasks, the in-context samples are composed of coordinates, without extra high-level semantic information. In this case, LLMs have to identify the patterns behind these two sets of data and predict the labels of query data accordingly. Thus, **the ML classification task is a more challenging benchmark for evaluating the in-context learning capability of LLMs**.
>
> **_[Task Settings.]_** The tasks adopted in our paper are the simple ML classification tasks. Different from conventional ML classification tasks, the test data, which are derived from the plane where the in-context samples lie, of the classification tasks adopted in our paper are not labeled. In such a case, **query data contains both in-distribution (ID) data, which lie near the in-context samples, and out-of-distribution (OOD) data, which is relatively far away from in-context data.  From this perspective, the tasks adopted in our work simulate the environments of the open-world/real-world tasks.**
>
> **_[High-dimension Issue.]_** In the context of the task settings mentioned above, **the main reasons that impede us from validating our findings in the high-dimensional case are (1) the lack of labels and (2) the exponential increase of query samples**. Specifically, on the one hand, it is hard to evaluate the decision boundaries in the higher dimension due to the lack of labels. To address this issue, we propose "smooth score" in our paper to quantitatively evaluate the smoothness of the decision boundary. However, due to the k-NN-based metric in the smooth score, it cannot be simply applied in the higher dimension. On the other hand, since the query data are generated directly from the plane/space where the in-context samples lie, increasing the dimension will increase the number of query data exponentially. Meanwhile, with the increase of dimension, the prompts will become longer. All these aspects will result in huge computational overhead. To address all these concerns simultaneously, we propose to evaluate LLMs on 3D tasks as a proxy of high-dimensional tasks. According to the results, the findings are consistent.
>
> **_[Multi-class Settings.]_** Per your concern, we are preparing the experiments. We will provide the results in the updated version of our paper once the experiments are completed.
>
> > Q1. 1. How is the validity of the "Smooth Score" verified? If a model (LLM) outputs a "trivial solution," such as predicting the same class (e.g., all 0s) for all query points, its "Smooth Score" would be a perfect 1.0.
>
> **Response:** Thank you for your question. Here, we would like to first clarify that **accuracy is NOT equivalent to smoothness**. In the case where the smooth score $s=1$, all query samples are assigned with the same label. Thus, the decision boundary is smooth. This is a desirable advantage of our proposed smooth score.
>
> > Q2. 1. The "KL-divergence-based statistical method" should report variance (or standard deviations) for its averaged results, in order to demonstrate whether the "gaps" (which the paper relies on for comparison) are statistically significant.
>
> **Response:** Thank you for your constructive and insightful suggestions. We will add variance/standard variance in the updated version of our paper.

---

### Official Review · Reviewer_QQyP · 2025-11-01

**Soundness:** 3
**Presentation:** 3
**Contribution:** 3
**Rating:** 4
**Confidence:** 3

**Summary:**

The paper probes in-context learning (ICL) behavior of LLMs on simple ML classification tasks (linear, circles, moons; plus a 3-D extension). It compares implicit vs explicit reasoning prompts, examines what happens when prompts explicitly name ML strategies (e.g., “use k-NN/SVM/DT/MLP”), and asks whether LLM decisions resemble those of conventional ML. Core findings: (i) LLM decision boundaries are fragmented, and implicit prompting yields less fragmented boundaries than explicit CoT; (ii) naming ML strategies tends to worsen boundary smoothness; (iii) using a KL-divergence gap test, the authors find that under a standard (implicit) prompt, LLM outputs on linear tasks are closest in distribution to conventional ML; (iv) distribution shift matters, when query and in-context data are aligned (ID queries), LLMs look much more like conventional ML; when the grid includes OOD regions, fragmentation grows. They also propose a quantitative Smooth Score (local k-NN agreement) to study smoothness beyond 2-D.

**Strengths:**

- The same two effects show up across several models and are interesting: decision boundaries from LLMs often look fragmented, and implicit prompting gives cleaner maps than explicit prompting (shown first on Llama-3-8B and replicated on DeepSeek-V3, Mistral-8B, and Qwen-2.5-7B).
- The explicit-reasoning examples are informative: the paper documents that models often describe their process in ML-like terms (e.g., analyzing patterns or invoking k-NN/DT), which is a nice qualitative complement

**Weaknesses:**

- Why we need to study LLM behaviours on linear classification tasks? The paper centers its analysis on toy ML classification tasks (linear / simple geometric datasets). Even the authors position these as proxies “to dive into the behaviors of LLMs,” not as tasks that matter in practice, where conventional ML already yields accurate, fast, and cheap solutions. The work does not articulate why understanding LLM behavior on these simplified settings will translate to real downstream wins for LLM prompting or evaluation.
- The proposed metric (“smooth score”) may lack some analyses. Smooth score depends on K-NN neighborhoods, but the paper does not study sensitivity to K, nor relate the score to other boundary complexity measures.
- A central result is that specifying ML strategies in the prompt degrades decision-boundary smoothness. But the study does not isolate whether this comes from instruction-following overhead, longer context length, token budget, or spurious tool-use emulation. A controlled ablation might help.
- Evidence for “LLMs implicitly leverage ML algorithms” is not strong enough. Stronger causal probes (interventions, counterfactual prompts, controlled transforms) might help.

**Questions:**

- The finding that implicit reasoning yields smoother boundaries is interesting, but the explanation (extra instructions “damage capability”) is conjectural without mechanistic analysis, could author further explain why implicit reasoning has smoother boundaries?

---

> ### Author Response · Authors · 2025-11-27
>
> > W1. Why we need to study LLM behaviours on linear classification tasks? The paper centers its analysis on toy ML classification tasks (linear / simple geometric datasets). Even the authors position these as proxies “to dive into the behaviors of LLMs,” not as tasks that matter in practice, where conventional ML already yields accurate, fast, and cheap solutions. The work does not articulate why understanding LLM behavior on these simplified settings will translate to real downstream wins for LLM prompting or evaluation.
>
> **Response:** Thank you for your comments. According to your comments, we notice that your main concern here is the significance of evaluating LLMs on simple ML classification tasks since more accurate, fast, and cheap solutions are available.
>
> As we have mentioned in the paper, **the main goal of evaluating LLMs on these simple ML classification tasks is to understand the behaviors of LLMs in in-context learning on more abstract and complex reasoning tasks**.
>
> Compared to conventional reasoning tasks, such as math reasoning, **performing reasoning on these classification tasks helps evaluate the true reasoning capability of LLMs**. Specifically, in our paper, we transfer the conventional ML classification tasks to reasoning tasks by describing the tasks with texts instead of matrices or vectors. In such reasoning tasks, the in-context samples are composed of coordinates, without extra high-level semantic information. In this case, LLMs have to identify the patterns behind these two sets of data and predict the labels of query data accordingly. Thus, **the ML classification task is a more challenging benchmark for evaluating the in-context learning capability of LLMs**.
>
> In our paper, we find that **(1) LLMs with implicit reasoning perform better than those with explicit reasoning**, **(2) LLMs with implicit reasoning behave in a similar way to conventional machine learning methods**, and **(3) LLMs with explicit reasoning show high perplexity (uncertainty) in the predictions, and majority voting can partially mitigate the fragmented decision boundaries and improve the perplexity**. All these findings provide new perspectives to study the in-context learning capability of LLMs and inspirations for performing tasks similar to machine learning classification tasks.
>
> > W2. The proposed metric (“smooth score”) may lack some analyses. Smooth score depends on K-NN neighborhoods, but the paper does not study sensitivity to K, nor relate the score to other boundary complexity measures.
>
> **Response:** Thanks for your comments. According to your comments, we notice that your main concern here is that the sensitivity to K of our proposed smooth score is not well explored.
>
> In fact, **we have conducted corresponding studies in our paper, and the analyses are reported in Appendix E**. Specifically, we calculate the smooth scores of the prediction results of Llama-3-8B on the 2D linear classification task under the settings of implicit reasoning, respectively, with $k = {3, 4, 10, 100}$. According to the empirical results, we find that the smooth scores of decision boundaries obtained from the sklearn tools are higher than those derived from LLMs, which is consistent with the visualization results that sklearn achieves better decision boundaries than LLMs. Moreover, we can also observe that the decision boundary generated from the standard settings achieves a better smooth score than the cases where machine learning strategies are specified when k is not so large. However, when k is quite large (e.g., 100), the case changes. Thus, we can summarize that the smooth score is robust to k when k varies in a reasonable range.
>
> A concern regarding our proposed smooth score is that the nearest-neighbor signals fade quickly when the dimension of the space increases. Moreover, we also notice that increasing the space dimension will result in the exponential increase of the amount of query data, which, in turn, increases the cost of reasoning. Thus, as a trade-off, we conduct our high-dimensional experiment in 3D space. Thus, a potential limitation of the smooth score is the space dimension. Specifically, the proposed smooth score cannot be applied to very high dimensions.

---

> ### Author Response · Authors · 2025-11-27
>
> > W3. A central result is that specifying ML strategies in the prompt degrades decision-boundary smoothness. But the study does not isolate whether this comes from instruction-following overhead, longer context length, token budget, or spurious tool-use emulation. A controlled ablation might help.
>
> **Response:** Thanks for your comments. According to your comments, we notice that your main concern here is the lack of ablation studies on factors that may affect the performance. Actually, some experiments in our paper may help address your concern.
>
> **_[Instruction-following Overhead.]_** According to your concern about instruction-following overhead, in **Appendix G**, we have provided both prompts adopted in the implicit and explicit reasoning. The main differences between the two prompts are whether the LLMs are allowed to output reasoning processes and whether the reasoning strategies are specified. We obtain the following two observations. **(1) Under the implicit reasoning settings, we conducted experiments with and without specified ML methods (Figs. 1(e), 1(f), & 3). With the empirical results, we can observe that specifying ML methods does not significantly modify the decision boundaries.** **(2) However, with the standard prompt, when LLMs are allowed to output reasoning processes, the decision boundaries degrade significantly.** All these results demonstrate that the explicit reasoning undermines the performance.
>
> **_[Longer Context Length.]_** According to your concern, a feasible way to determine the effect of context length is to perform an ablation study on the number of in-context samples. This has been explored in Zhao et al. 2024 (Section 4.2). Although the accuracy increases, the decision boundaries remain fragmented.
>
> **_[Token Budget.]_** According to your concern, we agree that the token budget plays an important role in reasoning tasks. The incomplete reasoning may negatively affect the performance. In our paper, for the implicit reasoning, the token budget is always sufficient since only the prediction and the confidence are output. For the explicit reasoning, we have set the token length to the maximum of the LLMs. Meanwhile, since answers can be extracted from the outputs, the reasoning is the completed output. With all these aspects taken into consideration, we can summarize that the token budget is not a potential reason for the fragmented decision boundaries.
>
> **_[Spurious Tool Invocation.]_** According to your concern about tool invocation of LLMs, we actually did not notice any changes in system processing and memory. Meanwhile, we noticed that LLMs tend to perform the selected algorithms during their reasoning process. Thus, we think that the LLMs did not explicitly invoke the tools in the reasoning.
>
> - Zhao et al., Probing the decision boundaries of in-context learning in large language models. NeurIPS 2024.
>
> > W4. Evidence for “LLMs implicitly leverage ML algorithms” is not strong enough. Stronger causal probes (interventions, counterfactual prompts, controlled transforms) might help.
>
> **Response:** Thanks for your suggestion. We agree that causal tools can potentially contribute to understanding the behaviors of LLMs. We will try to conduct corresponding experiments.
>
> > Q1. The finding that implicit reasoning yields smoother boundaries is interesting, but the explanation (extra instructions “damage capability”) is conjectural without mechanistic analysis, could author further explain why implicit reasoning has smoother boundaries?
>
> **Response:** Thanks for your question. This is an interesting and insightful question. We can start from the results of implicit and explicit reasoning. Compared to implicit reasoning, the decision boundaries derived from LLMs are more fragmented. However, the main difference between the two kinds of reasoning paradigms is whether reasoning steps are allowed to be output. Since the final answers are generated in the auto-regressive way, the final answers derived from explicit reasoning may be affect by the reasoning steps. This can be further demonstrated by the perplexity of explicit reasoning. Compared to implicit reasoning, the perplexity of each query sample is high, which indicates high uncertainty. Thus, **we can conjecture that the LLMs inherently owns the capability of handling ML classification tasks (implicit reasoning), while uncertainty in reasoning will damage such a capability**.

---

### Meta-Review · Area_Chair_vu1Q · 2025-12-30

**Summary:**

This paper investigates the ICL capability of LLMs by looking at decision boundaries on simple machine learning tasks (on simple synthetic datasets). The main observation is that LLMs fail to achieve a smooth decision boundary (measured by a proposed smooth score) and implicit reasoning achieves better decision boundaries than explicit reasoning. While the reviewers found the high level message to be interesting, there were many concerns on various aspects of the work, including the size of the model in the experiments, the choice of tasks, theoretical understanding, proposed smooth score, and the claim of "LLMs tend to address classification tasks in the way of machine learning algorithms". Some of these concerns were addressed by the revision (such as size of model) and some of them are partially addressed by author response. Overall, the reviews are fairly consistent with each other and some of the questions are only partially addressed.

**Reviewer Concerns:**

1. Only on models of size at most 8B: this is mostly addressed in the revision.
2. Concerns on the choice of tasks: the authors discussed why they believe their choice is reasonable. While there is certainly some merit I don't feel they addressed concerns of everyone.
3. theoretical understanding of smooth score: some concerns are addressed (some are mostly misunderstanding or different priorities), while there is also no strong theoretical foundation.
4. Need convincing evidence for "specifying ML strategies in the prompt degrades decision-boundary smoothness" and "LLMs tend to address classification tasks in the way of machine learning algorithms". Author response pointed to some of the experiments which partially addresses these concern, but they are not completely convincing.

**Reviewer Scores:**

QQyP: points partially addressed. Likely to stay at 4
nLo7: some points are well addressed. Likely to increase to 5
Tr9z: concern about model size addressed, others are up to discussion. Likely to stay at 4.
ev9p: points partially addressed. Likely to stay at 4

---

### Decision · Program_Chairs · 2026-01-26

Reject